**Data Availability Statement:** Data cannot be shared publicly because 'data sharing' was not

# The impact of comorbid spinal pain in depression on work participation and clinical remission following brief or short psychotherapy. Secondary analysis of a randomized controlled trial with two-year follow-up

**Marjon E. A. Wormgoor**[1,2]*, **Aage Indahl**[2], **Jens Egeland**[1,3]

**1** Division of Mental Health & Addiction, Vestfold Hospital Trust, Tønsberg, Norway, **2** Division Physical Medicine and Rehabilitation, Vestfold Hospital Trust, Stavern, Norway, **3** Department of Psychology, University of Oslo, Oslo, Norway

* marjon.wormgoor@siv.no

## Abstract

### Objectives

This explorative study analyses the influence of baseline comorbid long-lasting spinal pain (CSP) on improvement of long term work participation and clinical remission of mental health illness following either brief coping-focussed or short-term psychotherapy for depression. Whether type of treatment modifies outcome with or without CSP is also analysed.

### Design

A secondary post hoc subgroup analysis of a pragmatic randomised controlled trial.

### Interventions

Brief or standard short psychotherapy.

### Methods

Based on baseline assessment, the sample was subdivided into a subgroup with and a subgroup without CSP. Work participation and clinical remission of depression and anxiety were assessed as treatment outcome at two-year follow-up. Simple and multivariate logistic regression analyses, across the intervention arms, were applied to evaluate the impact of CSP on treatment outcome. Selected baseline variables were considered as potential confounders and included as variates if relevant. The modifying effect of CSP on treatment outcome was evaluated by including intervention modality as an interaction term.

included in the written consent form. Data are sensitive in nature and, as such, availability is restricted and regulated by Norwegian Laws and EC laws (GDPR). Application for access to anonymized data files can be sent to Vestfold Hospital Trust by e-mail to Forskning@siv.no.

**Funding:** The author(s) received no specific funding for this work.

**Competing interests:** The authors have declared that no competing interests exist.

## Main results

Among the 236 participants with depressive symptoms, 83 participants (35%) were identified with CSP. In simple logistic regression analysis, CSP reduced improvements on both work participation and clinical remission rate. In the multivariate analysis however, the impact of CSP on work participation and on clinical remission were not significant after adjusting for confounding variables. Reduction of work participation was mainly explained by the higher age of the CSP participants and the reduced clinical remission by the additional co-occurrence of anxiety symptoms at baseline. The occurrence of CSP at baseline did not modify long term outcome of brief compared to short psychotherapy.

## Conclusions

CSP at baseline reduced work participation and worsened remission of mental health symptoms two-year following psychotherapy. Older age and more severe baseline anxiety are associated to reduced effectiveness. Type of psychotherapy received did not contribute to differences.

## Introduction

In the general population, common mental health and musculoskeletal complaints are the leading causes of non-fatal health loss, reduced quality of life and work absence [1, 2]. These health complaints are frequently co-existing [3–9] and their shared biological pathways and symptoms has been pointed out frequently [4, 10]. It has been suggested that a bidirectional relationship exists between chronic spinal pain and depression, in which spinal pain is a risk factor for depression and depression is a risk factor for spinal pain [10]. Previous research has suggested that comorbidity is associated with additive or synergistic effects of increased symptom severity and disability and a negative impact on clinical remission and quality of life outcomes following intervention [4, 11–16]. Focus on the impact of comorbid pain on treatment effectiveness of depression, has been scarce [4, 7]. Research, however, has mostly focused on the impact of comorbid depression in non-specific low back pain (LBP) interventions. It has been suggested that comorbid depression might have an adverse effect on the prognosis and treatment effectiveness of LBP [17]. Evidence of a negative impact of the co-existence of depression and chronic LBP on return to work is still insufficient [17–20].

Previous reviews showed that brief interventions may be effective with regard to clinical outcome, in participants with common mental health problems [21–24], as well as in LBP [25, 26]. Even though many studies have favoured short term psychotherapies above brief psychotherapies for depression, we found in a recent pragmatic randomized controlled trial (RCT), that the briefest intervention was superior in enhancing early work participation and long term clinical remission [27]. In the brief psychotherapy, the focus was on normalizing, accepting and coping with present mental health complaints and hindrance for work participation. In the other intervention, short-term psychotherapy, there was in addition to the themes of the brief intervention focus on processing other challenging issues and previous pathogenic experiences.

Insight into the impact of CSP in psychotherapy for depression and on the interaction with the efficacy of different psychotherapy approaches can be clinically useful for the expectations of intervention efficacy and to guide clinical decision making for this substantial subgroup

with comorbid spinal pain [28]. Although brief psychotherapy is effective in a heterogeneous population, different intervention responses may occur in subgroups. It is important to identify the subgroups for which brief psychotherapy may be insufficient. A considerable part of the participants in our previous study reported both depression and back or neck complaints at baseline. As it has been suggested that somatic comorbidity entails usually an increased number of sessions [4], brief psychotherapy might be insufficient for the subgroup with comorbid CSP. As far as we know, comorbid long-lasting spinal pain (CSP) has not yet been studied as a moderator of differential treatment response in depressive participants of brief or standard short term psychotherapy. Short term psychotherapy allows for greater focus on previous or current challenging issues, including work and additional pain issues and may therefore be more beneficial for the participants with comorbid CSP complaints than brief psychotherapy.

The main research question of the current study is to determine whether baseline CSP is prognostic for long-term sick leave and reduced clinical remission of mental health problems following psychotherapy for depression. Second, it is questioned whether CSP may moderate the effect of brief versus short psychotherapy. To answer the first question, we will evaluate whether baseline CSP affects improvement rates on sick leave and clinical remission of mental health problems two-year following psychotherapy for depression and to estimate the magnitude of that effect, if any. To enable adjustment for confounding, we need to examine the differential associations between CSP and other baseline characteristics and their association with intervention outcome as well. Secondly, we will evaluate whether the effects of brief-coping focused and short psychotherapy with a more extended focus are equally beneficial in participants with baseline CSP compared to participants without CSP, or not.

## Methods

### Study design and setting

This study is a non-prespecified secondary analysis applying data from a pragmatic randomized controlled trial (RCT) with two-year follow-up. For more detailed description of methods and progress of the participants through the trial, see Wormgoor et al. [27] and Fig 1. The study was conducted at a transdiagnostic outpatient clinic for individuals with mental health or musculoskeletal complaints. In order to improve generalisability of findings, a pragmatic design was chosen, following ordinary clinical procedures with respect to patient inclusion, participation in minor additional treatment modalities and decision of therapy termination. The RCT assessed long-term effectiveness on WP and mental health of brief coping focused psychotherapy (Brief-PsT) and short term psychotherapy of standard duration with more extended focus (Short-PsT) for patients with primarily moderate depression or anxiety complaints.

Prior to any intervention, a questionnaire package containing various standardised validated Norwegian versions of questionnaires, was completed at the clinic. For follow-up, a questionnaire was sent by mail with reply-envelope at two-year following baseline.

Randomization procedure was concealed and based on computer generated randomization lists but stratified by gender. The participants were further stratified into two parallel trials according to self-rated symptom severity (mild vs. substantial). The participants were categorized as having 'substantial mental health complaints' (73% of all participants) if they classified their complaints as 'some' (score 2) or 'seriously affected' (score 3) on at least one of the anxiety or depression items (#28 and #29) of the Subjective Health Complaint inventory (SHC) [29]. In the current and the primary study, only data from the participants with substantial mental health complaints were applied.

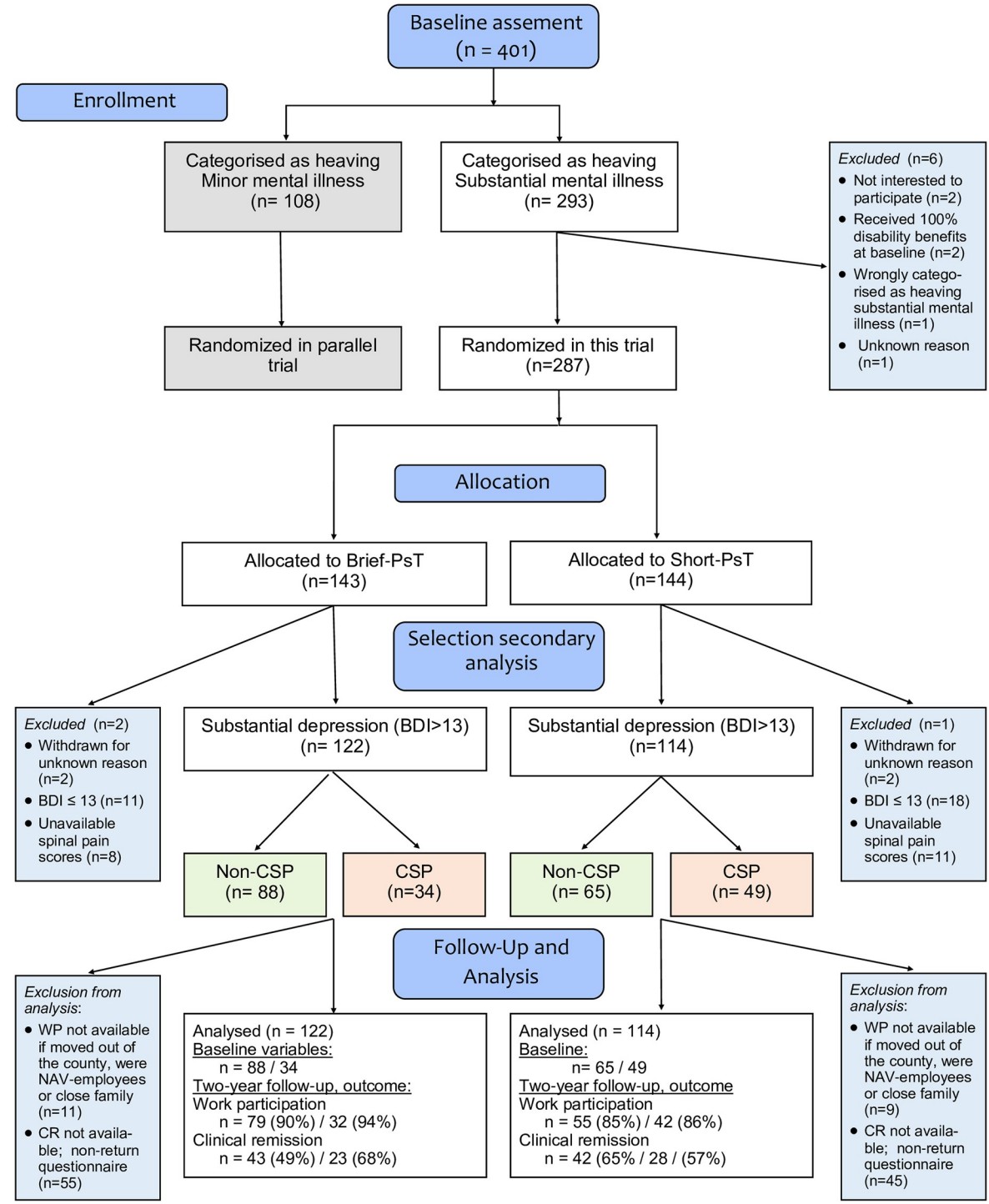

**Fig 1. Flowchart.** The initial screening procedure has been reported in a previous publication [27]; CSP-Comorbid Spinal Pain; BDI- Beck Depression Inventory-II.

## Participants

The RCT followed the ordinary outpatient clinic routine criteria and enrolled patients, aged between 18 and 65 years, who were on or at risk of sick leave, primarily due to common mental

health disorders. Sick leave had to be less than nine months during the preceding two years. All referrals were assessed by the clinic's psychologist coordinator. Participants were included to the clinic irrespective of comorbidities, but in cases of acute or more severe psychological pathology that required greater input than the outpatient clinic could offer, the patient was excluded and referred to other psychiatric service.

For the analyses in the present study the participants were included if they had obtained a baseline score >13 on the Beck Depression Inventory-II (BDI-II) [30] indicating mild to severe depression (recall period of 2 weeks) [31]. In addition they should have classified their mental health symptoms as substantial as described above. All participants signed informed consent prior to their participation.

The participants were divided in two subgroups, based on the presence of CSP. This was assessed with two questions of the baseline questionnaire. The duration of the pain was assessed with the single question: "Have you had musculoskeletal complaints for at least three months in a row? (by musculoskeletal disorders, we mean pain in muscles and joints, such as the neck, back, seat, hip, pelvis, leg, foot, shoulder or arm)" Answering options were 1. "I have never had musculoskeletal complaints for three months or more", 2. "I have been bothered for three months or more, but not right now" 3. "I'm currently bothered". The presence of sub-stantial spinal pain (SP) was assessed with the Subjective Health Complaint inventory (SHC) [29]. Neck, lower or upper back pain complaints (item #5–7) graded as 'some' (score 2) or 'seriously affected' (score 3) during the last 30 days, were considered as SP. The participants that fulfilled both criteria, musculoskeletal pain in the last three months with SP during at least the previous 30 days, were categorized as 'CSP'. The other subgroup without CSP was labelled as 'non-CSP'

## Intervention

The study was incorporated into the usual healthcare setting. Their general approach was the assumption that subjective health complaints are inevitable features of human life and provid-ing the patients with coping skills to manage these. Before starting psychotherapy the patients participated in a 5-hours transdiagnostic group education. The education was aimed at provid-ing insight and understanding with common health complaints, including musculoskeletal pain, anxiety, depression, and stress.

Following the education, concurrently to the psychotherapy, all participants had the oppor-tunity to participate in a five-day coping course, individual coaching sessions with another health care professional or a clinical examination with educational approach with a physiotherapist.

The psychotherapy has been described elsewhere [27]. Summarized, the Brief-PsT psycho-therapy aimed at up to 6 sessions, with focus on normalizing, accepting and coping with their present mental health complaints and their hindrance for WP. In the Short-PsT, there was an additional emphasis on extensive anamnesis and the possibility to establish a so-called central theme based on previous or current challenging issues and deeper focus on cognitive maladap-tive coping strategies or dynamic repetitions. The number of sessions was aimed to be 12–20 on average.

## Measures

The primary outcome for the analyses were both full work participation and clinical remission of depressive and anxiety symptoms at two-year follow-up. Work participation state was assessed from sickness and disability benefit data obtained from registry data from the Norwe-gian Labour and Welfare Administration (NAV) for baseline, three-month and one and

two-year follow-up. Benefits <30% of ordinary working time at two-year follow-up were considered as full work participation (WP2yr). Clinical remission (CR2yr) was defined as obtaining scores below the clinical cut-off score of both BDI-II ($\leq$ 13) and Beck Anxiety Inventory (BAI, $\leq$ 9) obtained from the questionnaire at two-year follow-up [31, 32]. Substantial spinal pain at follow-up was evaluated as a secondary outcome measure.

A questionnaire package evaluated sociodemographic characteristics, clinical measures, health management measures and work-related factors at both baseline and follow-up.

*Sociodemographic characteristics* included gender, age, marital status and highest educational level completed.

**Clinical measures.** Severity of depression symptoms was evaluated with BDI-II [33]; sub-dimension scores (cognitive, affective and somatic) were evaluated according to a 3-factor BDI-II model [34]. Severity of anxiety was evaluated with the Beck Anxiety Inventory [35] (BAI). The 30-day prevalence of 29 comorbid subjective somatic and psychological symptoms was assessed with the SHC [29]. Higher scores on BDI-II and BAI indicates higher severity. In addition, prevalence of number of SHC besides spinal pain, depression or anxiety ('SHC-other', without items #5–7, 28 and 29) and prevalence of individual deviant substantial pain items (score 3 or 4) were reported. Severity of last week's mental health and musculoskeletal complaints respectively were assessed on an 11-point numeric rating scale (NRS). Duration of mental health problems, first-time episode with mental health problems lasting more than three months and received treatment for current complaints during the preceding two years and during the study were assessed by single questions.

**Health management measures.** Beliefs of coping with various demands in life, was assessed with the General Self-Efficacy scale [36] (GSE), with mean scores above 3.0 indicating high self-efficacy. Participants' perceptions about their health complaints were measured by the Brief Illness Perceptions Questionnaire (BIPQ) [37], a nine-item questionnaire that applies single-item 11-point NRS scales to assess cognitive representations, emotional representations and coherence or comprehensibility of their health complaint. Higher overall scores indicate more pessimistic representations of illness. Health complaints related beliefs towards work attendance were assessed with an adapted version (phrasing 'health complaints' instead of 'low back pain') of the Work subscale of the Fear-Avoidance Beliefs Questionnaire (FABQwrk). The FABQ, although originally developed for LBP, adapted versions has been evaluated and widely used for other populations and considered as a useful prognostic tool for individuals on sick leave due to both musculoskeletal and psychological disorders [38, 39]. Agreement with 7 statements on a 7-point Likert scale was evaluated; higher scores indicate more strongly held fear-avoidance beliefs. Item #12 "I should not do my normal work with my present health complaints" was evaluated separately. Quality of life was measured with the Life Satisfaction questionnaire [40] (Lisat-11) targeting various life domains on a 7-point Likert scales. Higher scores indicate a greater degree of life satisfaction. Satisfaction with physical and mental health (item 10 and 11) were evaluated additionally, dichotomized as dissatisfied (score 0–3) vs. satisfied (score 4–6).

**Work-related factors.** Sick leave during previous year because of current health problems (dichotomized as less or $\geq$ 100 days) and usual working time (daytime or other) work-related factors were assessed by single questions. Superior and co-worker support were assessed with the question "At work, do you have a good relationship with your superiors?" and "People at work understand that I may have a bad day" of the Swedish Demand Control Support Questionnaire [41] (#S4 and 5 DCSQ), dichotomized as yes (often) and no (never/ seldom/ sometimes). Workload was evaluated with the questions "My job is physically demanding" (#5, Effort Reward Imbalance questionnaire [42]) and "My job is mentally demanding", dichotomized as 'does not apply', or 'does apply'. Job satisfaction was assessed with the question "Do

you enjoy being at work". Workplace uncertainty was evaluated with question #59 of General Nordic Questionnaire for Psychological and Social Factors at Work (QPSNordic) [43] "Are there rumours concerning changes at your workplace?". Both questions were assessed at a 5-points Likert scale and dichotomized as 'no' (never/very seldom, rather seldom or sometimes) or 'yes' (rather often and very often/always).

With the exception of one omitted inventory (Hopkins Symptoms Checklist-10) giving information overlapping with the Beck-inventories and BIPQ that was added in these current analyses, the methods applied in the present study equals to those reported in Wormgoor [27]. The difference in methods being that we now look for prognostic and effect modifying impacts and divides the sample as to CSP or non-CSP.

## Statistics and data analysis

The dependent variables in this study are the binary indicators (yes/no) of treatment response regarding full WP at two-year follow-up (WP2yr) and clinical remission of mental health symptoms at two-year follow-up (CR2yr).

Potential confounding baseline variables were selected based on a review of the literature concerning work participation and clinical outcome in common mental health or spinal complaints (Table 1). In this context, confounding variables are both related to CSP and affect intervention outcome. The included baseline variables were subjected to descriptive statistical analysis stratified for the presence of CSP. Data of both intervention arms were pooled. Their relationship with both CSP and intervention outcome were evaluated as described below in the selection of covariates for the regression models. Categorical and ordinal data are presented as proportions (%), normally distributed numeric data as means with standard deviation (SD), and non-normally distributed data as medians with interquartile range (IQR). Shapiro-Wilk test was applied to determine normality of data. Pearson's Chi square test, Student t-test and Mann–Whitney U test for non-normally distributed continuous variables were applied to perform baseline subgroup comparisons of sociodemographic, clinical, health management and work-related measures and the obtained intervention.

WP time effects are operationalized as the nine possible transitions of WP-state (no, partial or full) from baseline to follow-up (see Wormgoor [27]). To test within-group intervention effects of WP-state, Wilcoxon sign rank test was applied. Overall subgroup differences were evaluated with Mann–Whitney U test. For between-group difference in CR rate at two-year follow-up, a Chi square test, with Yates' continuity correction, was performed.

To examine the prognostic and effect modifying impact of CSP, two separate models were tested, one for each treatment response variable: WP2yr and CR2yr. Each prediction models was developed in five consecutive steps. Step 1 involved selection of potential confounders as covariates, based on the results of the association of the initial selected baseline variables with CSP. To be comprehensive we chose a lenient level of differential significance for including possible covariates (p.15). In step 2, analyses (Chi-squared test with Yates' continuity correction for all 2x2 analyses and Mann–Whitney U test for numerical variables) were applied to assess the associations of these variables with baseline WP and treatment response for both defined binary dependent outcome variables: WP2yr and CR2yr. Variables with associations to outcome with $p < .15$ were further considered as potential covariates. In order to avoid collinearity and to decide for final selection of covariates, in step 3 Phi coefficients ($\varphi$) as measure of association for the selected independent variables were calculated as well. Variables with $\varphi > 0.4$ were considered for elimination.

Next, baseline CSP was tested as a prognostic predictor for treatment response. Both intervention groups were combined in the analyses and regarded as one cohort. In step 4 direct

**Table 1. Summary statistics of demographics, clinical characteristics, health management and work measures at baseline, stratified for the presence of CSP.**

| | Range of values | Missing data n/n[1] | Non-CSP n | Non-CSP %, Mean, Median | Non-CSP (95% CI/ IQR) | CSP n | CSP %, Mean, Median | CSP (95% CI/ IQR) | Subgroup difference p-value |
|---|---|---|---|---|---|---|---|---|---|
| | | | 153 | 64.8% | (58.4–70.9) | 83 | 35.2% | (29.1–41.6) | |
| *Demographic characteristics* | | | | | | | | | |
| Gender, female, %* [c] | | | 98 | 64.1% | (55.9–71.6) | 61 | 73.5% | (62.7–82.6) | .18 * |
| Age, years, *median* [b] | 19–64 | | | 38.5 | (31.9–48.7) | | 42.7 | (36.2–51.4) | .01 * |
| > 40 years, % [c] | | | 71 | 46.4% | (38.3–54.6) | 55 | 66.3% | (55.1–76.3) | .005 * |
| Education, % [c] | | 1/0 | | | | | | | .64 |
| Low (Primary/lower secondary) | | | 9 | 5.9% | (2.7–10.9) | 6 | 7.2% | (2.7–15.1) | |
| Medium (upper secondary/Vocational education) | | | 73 | 48.0% | (39.9–55.3 | 44 | 53.0% | (41.7–64.1) | |
| High (University/college) | | | 70 | 46.1% | (38.0–54.3) | 33 | 39.8% | (29.2–51.1) | |
| *Clinical characteristics* | | | | | | | | | |
| 30-days prevalence of substantial spinal pain (SP), % [c] | | | 59 | 38.8% | (31.0–46.6) | | 100% | (95.7–100) | < .001 |
| Current musculoskeletal complaints, | | 4/0 | | | | | | | < .001 |
| > 3-months duration, % [c] | | | 16 | 10.7% | (5.8–15.7) | | 100% | (95.7–100) | |
| Current episode mental health problems, | | 3/3 | | | | | | | .33 |
| > 6 months duration, % [c] | | | 80 | 53.3% | (45.0–61.5) | 48 | 60.0% | (48.4–70.8) | |
| First-time episode with mental health problems, % [c] | | 4/0 | 68 | 45.6% | (37.5–53.6) | 27 | 32.5% | (22.7–42.6) | .07 * |
| Depression (BDI-II), *median sum score* [b] | 0–63 | | | 26 | (21–32) | | 27 | (22–32) | .84 |
| Mild depression (14–19) | | | 31 | 20.3% | (14.2–27.5) | 15 | 18.1% | (10.5–28.1) | |
| Moderate depression (20–28) | | | 61 | 39.9% | (32.1–48.1) | 36 | 43.4% | (32.5–54.7) | |
| Severe depression (29–63) | | | 61 | 39.9% | (32.1–48.1) | 32 | 38.6% | (28.1–49.0) | |
| BDI-II- cognitive dimension, *median mean score* [b] | 0–24 | | | 1.1 | (0.8–1.5) | | 1.0 | (0.9–1.5) | 1.0 |
| BDI-II- affective dimension, *median mean score* [b] | 0–24 | | | 1.3 | (1.0–1.6) | | 1.3 | (1.0–1.5) | .92 |
| BDI-II- somatic dimension, *median mean score* [b] | 0–15 | | | 1.4 | (1.2–1.8) | | 1.6 | (1.2–2.0) | .31 |
| Anxiety (BAI), median sum score [b] | 0–63 | | | 14 | (7–21) | | 17 | (11–25) | .01 * |
| Minimal anxiety (0–9) [c] | | | 52 | 34.0% | (26.5–41.5) | 18 | 21.7% | (13.4–32.1) | .12 * |
| Mild anxiety (10–16) | | | 42 | 27.5% | (20.6–35.2) | 21 | 25.3% | (16.4–36.0) | |
| Moderate anxiety (17–29) | | | 42 | 27.5% | (20.6–35.2) | 29 | 34.9% | (24.7–45.2) | |
| Severe anxiety (30–63) | | | 17 | 11.1% | (6.1–16.1) | 15 | 18.1% | (10.5–26.4) | |
| Mental health complaints, severity last week, *median* [b] | 0–10 | | | 7 | (6–8) | | 7 | (5–8) | .10 * |
| Subjective health complaints (SHC), #, *median* [b] | 0–29 | | | 12 | (9–15) | | 17 | (14–20) | < .001 * |
| SHC, substantial spinal pain (score ≥ 2), % [c] | | | 59 | 38.6% | (30.8–46.8) | 83 | 100% | (95.7–100) | < .001 |
| SHC, substantial insomnia (score ≥ 2), % [c] | | 1/0 | 113 | 74.3% | (66.6–81.3) | 67 | 80.7% | (70.6–88.6) | .35 |
| SHC–other complaints, #, *median* [b] | 0–23 | | | 8 | (6–11) | | 12 | (9–14) | < .001 * |
| *Health management measures* | | | | | | | | | |
| Self-efficacy (GSE), overall mean score, *mean*[a] | 1–4 | 0/1 | | 2.5 | (0.5) | | 2.6 | (0.5) | .58 |
| > *median* = 3, % [c] | | 0/1 | 30 | 19.6% | (13.3–25.9) | 14 | 17.1% | (9.7–27.0) | .64 |
| Illness Perception (BIPQ), overall mean score, *mean* [a] | 1–10 | | | 6.1 | (1.1) | | 6.3 | (0.9) | .08 * |
| 1. Consequences", *median* [b] | 1–10 | | | 8 | (7–9) | | 8 | (7–9) | .40 |
| 2. Timeline", *median* [b] | 1–10 | 3/2 | | 5 | (4–8) | | 7 | (5–8) | .001 * |
| 3. Personal control', *median* [b] | 1–10 | | | 3 | (2–5) | | 3 | (2–6) | .23 |
| 4. Treatment control', *median* [b] | 1–10 | 2/3 | | 7 | (6–9) | | 8 | (6–9) | .67 |
| 5. Identity", *median* [b] | 1–10 | 6/2 | | 5 | (3–7) | | 7 | (5–8) | < .001 * |
| 6. Concern", *median* [b] | 1–10 | 1/0 | | 8 | (7–9) | | 8 | (7–9) | .90 |
| 7. Understanding', *median* [b] | 1–10 | | | 7 | (4–8) | | 7 | (5–8) | .92 |
| 8. Emotional response", *median* [b] | 1–10 | 0/1 | | 9 | (8–10) | | 9 | (8–10) | .29 |

*(Continued)*

**Table 1.** (Continued)

| | Range of values | Missing data n/n[1] | n | % , Mean, Median | (95% CI/ IQR) | n | % , Mean, Median | (95% CI/ IQR) | Subgroup difference p-value |
|---|---|---|---|---|---|---|---|---|---|
| | | | **Non-CSP** | | | **CSP** | | | |
| Fear-avoidance Beliefs related to work (FABQwrk), overall mean score, *median* [b] | 0–6 | | | 3.0 | (1.5–4.1) | | 2.7 | (1.1–3.9) | *.35* |
| Should not do my normal work with my present health complaints, item 12, *median* [b] | 0–6 | 4/3 | | 3 | (1–5) | | 3 | (0–5) | *.23* |
| Life satisfaction (LISAT) mean score, *median* [b] | 1–6 | | | 3.7 | (3.2–4.1) | | 3.5 | (3.1–4.0) | *.12* * |
| 10. Dissatisfaction physical health ≤ 3, % [c] | | | 37 | 24.2% | (17.6–31.0) | 59 | 71.1% | (60.1–80.5) | *< .001* * |
| 11. Dissatisfaction mental health ≤ 3, % [c] | | | 139 | 90.9% | (86.3–95.4) | 76 | 91.6% | (83.4–96.5) | *1.00* |
| *Work characteristics* | | | | | | | | | |
| Sick leave previous year due to current health problems, % [c] | | | | | | | | | *.003* * |
| < 24 days | | 8/5 | 60 | 41.4% | (33.3–49.9) | 27 | 34.6% | (24.2–46.2) | |
| > 25–99 days | | | 71 | 49.0% | (40.8–57.1) | 30 | 38.5% | (27.7–50.2) | |
| ≥ 100 days | | | 14 | 9.7% | (5.4–15.7) | 21 | 26.9% | (17.5–38.2) | |
| WP expectations, *within some days or weeks*, % [c] | | 6/1 | 80 | 54.4% | (46.4–62.5) | 49 | 59.8% | (48.3–70.4) | *.52* |
| Normal daytime work, % [c] | | 5/1 | 101 | 68.2% | (60.1–75.7) | 60 | 73.2% | (62.2–82.4) | *.53* |
| Experienced distressing physical workload, % [c] | | 6/1 | 38 | 25.9% | (18.8–32.9) | 35 | 42.7% | (31.8–54.1) | *.01* * |
| Experienced distressing mental workload, % [c] | | 5/1 | 112 | 75.7% | (68.0–82.4) | 70 | 85.4% | (75.8–92.2) | *.12* * |
| Job satisfaction (often or always), % [c] | | 5/1 | 78 | 52.7% | (44.3–61.0) | 55 | 67.1% | (55.8–77.1) | *.048* * |
| Superior support (yes, often), % [c] | | 10/4 | 70 | 49.0% | | 45 | 57.0% | (45.3–68.1) | *.32* |
| Coworker support (yes, often), % [c] | | 6/4 | 123 | 34.7% | | 27 | 34.2% | (23.9–45.7) | *.94* |
| Rumours workplace uncertainty (sometimes/often), % [c] | | 6/2 | 32 | 21.9% | | 22 | 27.2% | (17.8–38.2) | *.37* |

0/1: score possibilities, 0 = negative and 1 = positive or true result

[1] Number of missing data analysed if departing from full dataset (n = 153 for non-CSP / n = 83 for CSP).

[a] Student t-test,

[b] Mann–Whitney U test,

[c] Pearson Chi-Square with Yates' continuity correction

' Lower scores indicated more negative perceptions;" Higher scores indicated more negative perceptions

* Selected as a potential confounder and covariate, step 1

binary logistic regression was performed to assess the impact of CSP alone (simple logistic regression). Step 5 involved direct multivariate logistic regression analyses in with CSP and the relevant covariates (from step 2) were forced in the regression model as one block. Crude and adjusted odds ratios (cOR and aOR) and corresponding 95% confidence intervals (95%CI) are reported for both treatment response variables. Hence OR >1.0 indicate that variable is associated with higher probability of response whereas OR <1.0 indicates an association with lower probability of response. In the final step, CSP was evaluated as a treatment effect modifier, i.e. baseline SCP and treatment response interacts with type of psychotherapy, treatment modality was included as an interaction term with CSP in a logistic regression analyses. Because of the presence of the interaction, treatment modality was centred and coded as -0.5 and 0.5 as recommended by Kraemer [44].

Little's missing completely at random test was applied to explore relevant baseline data. Subgroup differences in loss-to-follow-up for WP2yr and CR2yr were tested for all baseline variables. All analysis were performed by original assigned intervention groups. For the

presentation of WP-state at follow-up, full intention-to-treat (ITT) analyses were applied; the last-observation-carried-forward rule for missing data was used (last observation could be baseline, three months or one-year follow-up). For the analysis with WP2yr and CR2yr, the data were analysed as partial intention-to-treat analyses, excluding participants with no follow-up data. Sensitivity analyses were performed to see how redefining the threshold for subgroup defining changes the observed subgroup and intervention for differences for WP2yr and CR2yr. We repeated the analysis for four different thresholds: subgroups with the presence of only 1-month substantial spinal pain, of one-month any spinal pain, for one-month any spinal pain plus three months musculoskeletal pain or for only three months musculoskeletal pain.

Data analyses were performed using IBM SPSS Statistics for Windows, version 23 (IBM Corp., Armonk, N.Y., USA). A $p$-value of less than 0.05 was considered statistically significant.

### Ethics

All procedures followed were in accordance with the ethical standards of the responsible committee on human experimentation (institutional and national) and with the Helsinki Declaration of 1975, as revised in 2000. Informed consent was obtained from all individual participants included in the study. The original study design was registered and approved by the Regional Committee for Ethics in Medical Research in South-Norway (Ref. 2013/1034/REK Sør-ØstB) and registered at ClinicalTrials.Gov (ID: NCT04457635). Changes of the follow-up period (12 to 24 months) has been applied for and approved in 2016. The analysis as described in this current study had not been protocolled before the trial began. The idea of these secondary analyses was derived from interesting findings of the RCT [27] and was not foreseen by us as of potential significance before the study start.

## Results

Among the 236 participants with depressive symptoms included in these analyses, the 30-days prevalence of substantial neck, upper and lower back pain were 47%, 29% and 40% respectively. The aggregated prevalence of substantial spinal pain was 60.2%. For 42.7% of all participants, the duration of current musculoskeletal complaints was three months or more. The prevalence of CSP was assessed as 35.2%, comprising the participants that fulfilled both criteria. The 30-days prevalence of spinal pain in the non-CSP subgroup was 38.6% and of current musculoskeletal complaints (other than SP) with at least 3-months duration was 10.7%. We have performed sensitivity analyses to check whether other subgrouping thresholds effect treatment responses. For all variants, contrasts and level of significance between subgroup outcome of WP2yr and CR2yr was greatest in the subgrouping that is reported here.

WP2yr register data was available for 87.6% (non-CSP) and 89.2% (CSP) of the participants. CR2yr was calculated from BDI and BAI that were included in the questionnaire, but returned by only 55.6% and 61.4% respectively. Loss-to-follow-up was neither related to the presence of comorbid CSP nor to treatment allocation. Availability of WP2yr and CR2yr data were unrelated. Compared with those with complete data, participants with missing WP2yr data did not differ in baseline characteristics, nor in CR2yr. In both subgroups, participants with available CR-2yr status were more often female (61.2 vs 45.5%, $p = .09$ and 62.2% vs. 44.2%, $p = .04$) and older (median 42.7 vs 36.5, $p < .001$ and 45.6 vs 40.5, $p = .03$) than the non-repliers. In addition, CSP-repliers were more depressed (BDI-values, median 25.7 vs. 29.1, $p = .047$), reported lower life satisfaction (LISAT score 3.3 vs 3.6, $p = .009$) and lower BIPQ-timeline values (5.6 vs 6.3, $p = .046$). Further, non-CSP repliers were more often high educated (57.1% vs 32.4%, $p = .007$) and were more frequently satisfied with their job (61% vs 42.4%, $p = .04$). No other

significant baseline differences were found related to loss-to-follow-up of CR2yr. See Worm-goor [27] and flowchart in Fig 1. for further details about drop-out and loss-to-follow-up.

## Comparative characteristics according to the presence of comorbid spinal pain

To assess the balance between the groups at baseline with respect to the main prognostic or confounding factors, relevant baseline measures stratified for the presence of spinal pain (subgroup non-CSP vs subgroup CSP) are presented in Table 1. Missing values were, apart from the work characteristics items, close to zero percent and completely at random (Little's test, p = .24).

There were no differences in severity of depression, but patients with CSP showed a higher level of anxiety with a higher proportion with mild or severe anxiety. In addition to anxiety and SP, they reported a higher number of 30-days presence of several SHC's; in particular a higher prevalence of other substantial pain: arm (42.0% vs 14.8%, *< .001*), shoulder pain (68.3% vs 23.8%, *< .001*) and headache (69.9% vs 47.7% *p = .002*). In non-CSP the 30-days presence of substantial spinal pain was not correlated with the presence of clinical anxiety (BAI>9, φ = 0.09, *p = .03*). Anxiety and work-related fear avoidance beliefs were unrelated in both subgroups (φ = 0.20, *p = .01* in non-CSP and φ = 0.03, *p = .80* in CSP).

Regarding health management measures, CSP participants showed a more pessimistic belief of the duration of their health complaints (timeline) and attributed more symptoms to their health problem (identity).

At baseline assessment, the two subgroups showed comparable WP rates. However, in the previous year, CSP participants had more sick leave related to their current health problems. CSP participants experienced more often their job as physically demanding, yet a higher proportions indicated that they liked being at work.

## Comorbid spinal pain as prognostic factor of treatment response

Table 2 shows the treatment response measures for both subgroups. Irrespective of obtained intervention, WP rate and clinical scores increased significantly from baseline to two-year follow-up in both subgroups. At follow-up, full-WP was 85.0% in non-CSP and 65.1% in CSP (ITT analyses). CR2yr was reported as 67.1% and 43% respectively (partial ITT). However, both treatment response measures were significant lower in the participants with CSP. The odds ratio (OR) of positive WP2yr given the presence of CSP at baseline was 0.28 (95% CI: 0.13–0.58, *p < .001*) and for CR2yr 0.37 (95% CI: 0.18–0.76, *p = .007*). Notwithstanding any unsuccessful CR2yr, 88.0% of non-CSP and 52.0% of CSP had full-WP at follow-up (*p = .01*). Among the clinical responders, 92.5% of non-CSP and 89.5% had full-WP (*p = 1.00*). In non-CSP, WP2yr was unrelated to CR2yr status (φ = 0.07, *p = .53*) and in CSP, a moderate correlation was observed (φ = 0.40, *p = .007*).

In Table 3, subgroup prevalence of depression, anxiety and spinal pain at two-year follow-up are presented, as well as simultaneous occurrences of these symptoms. At follow-up, participants with baseline CSP reported more often clinical scores compared to non-CSP.

In both subgroups, having SP at follow-up was related to reduced CR2yr rates (non-CSP: 48.5% vs. 78.8%, *p = .008* and CSP: 32.4% vs. 71.4%, *p = .03*); baseline SP did not impact this (*p = .26* and *p = .82*). SP at follow-up did not affect WP2yr (non-CSP 90.9% vs. 91.3%, *p = 1.0*; CSP 90.6 vs. 60.6% *p = .14*). But, among the participants with follow-up spinal pain, WP2yr was significantly lower in CSP (p = .01).

The variables listed in Table 1 that revealed subgroup differences below *p = .15* (p-values marked with symbol *) were selected as potential confounding variables and therefore

**Table 2. Work participation degree at baseline and follow-up and clinical remission rate at follow-up, stratified for the presence of CSP.**

| | Non-CSP | | CSP | | Subgroup differences | | |
|---|---|---|---|---|---|---|---|
| | Total n | Positive treatment response | Total n | Positive treatment response | Effect size[e] | Test statistic | p-value |
| *Work participation at baseline* [a] | 153 | % | 83 | % | 0.10 | $\chi^2_{(1)} = 0.50$ | .481 |
| No-WP | | 57.5 | | 49.4 | | | |
| Partial-WP | | 18.3 | | 26.5 | | | |
| Full-WP | | 24.2 | | 24.1 | | | |
| *Work participation at two-year follow-up* [b] | 153 | | 83 | | 0.24 | $U_{(8)} = -4737$ | .001 |
| No-WP | | 12.4 | | 25.3 | | | |
| Partial-WP | | 2.6 | | 9.6 | | | |
| Full-WP | | 85.0 | | 65.1 | | | |
| *Time effects, test statistics* [c] | $Z_{(2)} = -8.64, p < .001$ | | $Z_{(2)} = -5.10, p < .001$ | | | | |
| *Clinical Remission at two-year follow-up* [d] | 85 | 67.1 | 51 | 43.1 | | $\chi^2_{(1)} = 6.54$ | .01 |

[a] Statistics for subgroup differences in WP state at baseline follow-up: Mantel-Hanzel Linear by Linear association ($\chi^2$)—ITT

[b] Subgroup differences in time effect of work participation (changes of WP-state from baseline to follow-up) were evaluated with Mann–Whitney U test ITT

[c] To test time effects of WP-state within each subgroup, Wilcoxon sign rank test was applied—ITT

[d] Statistics for subgroup differences in CR2yr are based on Chi-squared test with Yates' continuity correction—partial ITT analyses

[e] Effect sizes for work participation are reported as Cramer's V (V), for clinical remission as OR

subjected to analyses of associations to treatment response and baseline WP. The results are presented in Table 4. None of the potential confounders were associated with employment status at baseline. The baseline variables that were related to higher WP2YR or CR2yr respectively are marked with symbol * in Table 4. Phi calculations between the variables associated with treatment response showed no or weak association; LISAT total score and physical health score in addition to first-time episode and BIPQ-timeline showed the highest correlation (both φ = 0.40, p < .001), all other were below 0.30. Thus, collinearity between variables were not a significant dilemma for these potential confounders.

Direct logistic regression was performed with the variables which had shown an association with both CSP and the concerning treatment response. Results of the logistic regression

**Table 3. Prevalence's of clinical depression[a], anxiety[b] and spinal pain[c] and of simultaneous occurrences of symptoms at two-year follow-up.** Statistics are stratified for the presence of CSP.

| | Non-CSP (n = 85) | | CSP (n = 51) | | Subgroup differences | | |
|---|---|---|---|---|---|---|---|
| | % | (95% CI) | % | (95% CI) | OR | $\chi^2$ | p |
| Depression[a] | 28.2 | (19.0–39.0) | 47.1 | (32.9–61.5) | 2.26 | 4.16 | .04 |
| Anxiety[b] | 18.8 | (11.2–28.8) | 47.1 | (32.9–61.5) | 3.83 | 10.92 | .001 |
| Spinal pain[c] | 33.8 | (28.4–50.0) | 72.5 | (58.3–84.1) | 4.17 | 13.20 | < .001 |
| Depression & Anxiety | 14.1 | (7.5–23.4) | 37.3 | (24.1–51.9) | 3.61 | 8.43 | .004 |
| Depression & Spinal pain | 17.6 | (10.2–27.4) | 45.1 | (31.1–59.7) | 3.38 | 10.61 | .001 |
| Anxiety & Spinal pain | 10.6 | (5.0–19.2) | 39.2 | (28.8–53.9) | 5.45 | 13.91 | < .001 |
| Depression, Anxiety & Spinal pain | 8.2 | (3.4–16.2) | 35.3 | (22.4–49.9) | 6.08 | 13.80 | < .001 |
| No symptoms above threshold | 48.2 | (37.3–59.3) | 19.6 | (9.8–33.1) | 0.26 | 9.96 | .002 |

Statistics for subgroup differences of prevalence rates are evaluated with Chi-squared test with Yates' continuity correction.

[a] BDI>13,

[b] BAI>9,

[c] Substantial spinal pain during the last 30 days: SHC, score 2 or 3 on item #5, 6 or 7.

**Table 4. Association of the selected variables with treatment response (WP2yr and CR2yr) and baseline WP.**

| Dichotomous data [a] | | Work participation at baseline | | | | Work participation at two-year follow-up | | | | Clinical Remission at two-year follow-up | | | |
|---|---|---|---|---|---|---|---|---|---|---|---|---|---|
| | | Total N | % | $\chi^2_{(1)}$ | p | Total n | % | $\chi^2_{(1)}$ | p | Total n | % | $\chi^2_{(1)}$ | p |
| CSP | Non-CSP | 135 | 24.2 | 0.00 | 1.00 | 134 | 89.6 | 11.07 | .001* | 49 | 65.3 | 6.54 | .01* |
| | CSP | 83 | 24.1 | | | 74 | 70.3 | | | 87 | 54.0 | | |
| Gender | Male | 77 | 27.3 | 0.38 | .54 | 69 | 82.6 | 0.00 | 1.0 | 34 | 64.7 | 0.49 | .48 |
| | Female | 159 | 22.6 | | | 139 | 82.7 | | | 102 | 55.9 | | |
| First-time episode | no | 137 | 26.3 | 0.09 | .34 | 120 | 80.0 | 0.89 | .35 | 75 | 46.7 | 7.15 | .007* |
| | yes | 95 | 20.0 | | | 86 | 86.0 | | | 59 | 71.2 | | |
| Last year sick leave | < 100 days | 167 | 26.1 | 2.23 | .14 | 167 | 86.2 | 7.80 | .005* | 109 | 63.3 | 2.88 | .09* |
| | ≥ 100 days | 31 | 14.3 | | | 30 | 63.3 | | | 18 | 38.9 | | |
| Physical workload | no | 156 | 23.7 | 0.00 | 1.0 | 139 | 83.5 | 0.21 | .65 | 95 | 67.4 | 10.01 | .001* |
| | yes | 73 | 24.7 | | | 64 | 79.7 | | | 37 | 35.1 | | |
| Mental workload | no | 48 | 25.0 | 0.00 | .99 | 47 | 76.6 | 0.89 | .35 | 30 | 73.3 | 2.84 | .09* |
| | yes | 182 | 23.6 | | | 156 | 84.0 | | | 102 | 53.9 | | |
| Job satisfaction | no | 84 | 19.6 | 1.73 | .19 | 85 | 81.2 | 0.03 | .87 | 46 | 54.3 | 0.24 | .62 |
| | yes | 120 | 27.1 | | | 118 | 83.1 | | | 86 | 60.5 | | |
| Numerical data [b] | | WP-Median | WP +Median | U | p | WP-Median | WP +Median | U | p | CR-Median | CR +Median | U | p |
| Age, year | | 41.0 | 37.8 | 4205 | .046 | 40.9 | 39.2 | 2000 | .001* | 43.3 | 42.7 | 2107 | .52 |
| Anxiety (BAI), total score | | 15 | 14 | 4915 | .68 | 17.5 | 15.0 | 2678.5 | .20 | 22 | 12 | 1322.5 | < .001* |
| Mental health complaints, severity | | 7 | 7 | 5022 | .86 | 7 | 7 | 2783 | .33 | 7 | 7 | 1939.5 | .16 |
| Subjective health complaints -other # | | 14 | 14 | 4922 | .69 | 17.0 | 13.5 | 2426 | .04* | 15 | 14 | 1836.5 | .07* |
| Illness perception—timeline | | 6 | 7 | 4292 | .14 | 7 | 6 | 2220 | .02* | 6 | 5 | 1559.5 | .01* |
| Illness perception- identity | | 6 | 6 | 4461 | .335 | 7 | 6 | 2444.5 | .13* | 6.5 | 6 | 1885 | .43 |
| Life satisfaction (LISAT), mean score | | 3.6 | 3.6 | 5028.5 | .87 | 3.6 | 3.7 | 3023 | .08* | 3.5 | 3.8 | 1976 | .22 |
| Satisfaction with physical health | | 4 | 4 | 4701 | .36 | 3 | 4 | 2414.5 | .03* | 4 | 4 | 1764.5 | .03* |

Variables with associations to outcome with p < .15 were further considered as potential confounders.

* Selected as a covariate, step 2

At time of assessment: WP-: no full work participation; WP+: full work participation (= WP2yr); CR-: no clinical remission, CR+: clinical remission (= CR2yr)

[a] Pearson Chi-Square with Yates' continuity correction) was applied to assess the associations with the dichotomous baseline characteristics and WP, WP2yr and CR2yr

[b] Mann–Whitney U test was applied to assess the associations with numerical baseline characteristics and WP, WP2yr and CR2yr

analyses are presented in Table 5. Simple logistic regression with CSP as the only independent variable reduced WP2yr, but was not significant anymore when multivariate logistic regression was used to control for relevant covariates. Only age obtained significance (aOR = 0.93). For clinical remission, CSP alone produced a significant fit in the simple logistic regression analyses. After inclusion of the covariates, CSP was not significant anymore either and anxiety turned out as a significant negative effect modifier for CR2yr, with an aOR of 0.34. For both WP2yr and CR2yr, adding the relevant covariates increased the accuracy of the model compared to CSP alone.

**Table 5. Results of simple and multivariate logistic regression analysis for the association between CSP and associated variables with the dependent variables WP2yr and CR2yr.** In addition, the results of analysis of intervention interaction in the association of CSP with WP2yr and CR2yr.

| | OR | (95% CI) | Coef(B) | SE | Nagelkerke R² | Z | p | Model | |
| --- | --- | --- | --- | --- | --- | --- | --- | --- | --- |
| | | | | | | | | % correct | p |
| **WP2yr as dependent variable** | | | | | | | | | |
| *Simple logistic regression. n = 208* | | | | | 0.09 | | | 82.7 | .001 |
| CSP | 0.28 | (0.13–0.58) | -1.29 | 0.38 | | 11.49 | .001 | | |
| Intercept | 8.57 | | 2.15 | 0.28 | | 57.87 | < .001 | | |
| *Multiple logistic regression. n = 197* | | | | | 0.26 | | | 85.0 | < .001 |
| CSP | 0.51 | 1.78–1.47 | -0.67 | 0.54 | | 1.55 | .21 | | |
| Age | 0.93 | 0.89–0.97 | -0.08 | 0.02 | | 11.23 | .001 | | |
| Subjective health complaints-other | 0.99 | 0.89–1.10 | -0.01 | 0.05 | | 0.01 | .91 | | |
| Illness perception—timeline | 0.82 | 0.64–1.05 | -0.20 | 0.12 | | 2.55 | .11 | | |
| Illness perception—identity | 0.96 | 0.79–1.16 | -0.05 | 0.10 | | 0.20 | .66 | | |
| Last year sick leave | 0.43 | 0.15–1.20 | -0.85 | 0.53 | | 2.59 | .11 | | |
| Life satisfaction | 0.70 | 0.32–1.53 | -0.36 | 0.40 | | 0.82 | .37 | | |
| Satisfaction with physical health | 1.16 | 0.74–1.82 | 0.15 | 0.23 | | 0.44 | .51 | | |
| Intercept | 2390.3 | | 7.78 | 2.32 | | 11.22 | .001 | | |
| *Intervention interaction. n = 208* | | | | | 0.09 | | | 82.7 | .007 |
| CSP | 0.27 | (0.13–0.58) | -1.31 | 0.39 | | 11.17 | .001 | | |
| Intervention | 1.29 | (0.40–4.07) | 0.25 | 0.59 | | 0.18 | .67 | | |
| CSP * intervention | 0.68 | (0.15–3.14) | -0.39 | 0.78 | | 0.25 | .62 | | |
| Intercept | 7.78 | | 2.05 | 0.35 | | 33.56 | < .001 | | |
| **CR2yr as dependent variable** | | | | | | | | | |
| *Simple logistic regression. n = 136* | | | | | 0.31 | | | 63.2 | .001 |
| CSP | 0.37 | (0.18–0.76) | -0.99 | 0.37 | | 7.32 | .007 | | |
| Intercept | 2.04 | | -0.71 | 0.23 | | 9.49 | .002 | | |
| *Multiple logistic regression. n = 121* | | | | | | | | 71.1 | < .001 |
| CSP | 0.47 | 0.18–1.26 | -0.75 | 0.50 | | 2.23 | .14 | | |
| Anxiety | 0.93 | 0.89–0.98 | -0.07 | 0.03 | | 6.72 | .01 | | |
| First-time episode | 1.73 | 0.64–4.66 | 0.55 | 0.51 | | 1.17 | .28 | | |
| Subjective health complaints -other | 1.09 | 0.97–1.22 | 0.08 | 0.06 | | 2.01 | .16 | | |
| Illness perception—timeline | 0.47 | 0.77–1.26 | -0.02 | 0.13 | | 0.02 | .90 | | |
| Last year sick-leave ≥ 100 days | 0.94 | 0.11–1.52 | -0.88 | 0.66 | | 1.77 | .18 | | |
| Physical workload | 0.97 | 0.15–1.10 | -0.91 | 0.51 | | 3.15 | .08 | | |
| Mental workload | 0.69 | 0.22–2.16 | -0.38 | 0.59 | | 0.42 | .52 | | |
| Satisfaction physical health | 1.09 | 0.70–1.72 | 0.09 | 0.230 | | 0.16 | .69 | | |
| Intercept | 0.98 | | 1.17 | 1.87 | | 0.39 | .53 | | |
| *Intervention interaction. n = 136* | | | | | 0.12 | | | 64.0 | .007 |
| CSP | 0.37 | (0.18–0.77) | -0.99 | 0.37 | | 7.05 | .008 | | |
| Intervention | 0.40 | (0.16–1.03) | -0.91 | 0.48 | | 3.61 | .06 | | |
| CSP * intervention | 1.26 | (0.29–5.45) | 0.23 | 0.75 | | 0.10 | .78 | | |
| Intercept | 3.30 | | 1.19 | 10.94 | | 10.94 | .001 | | |

OR = Odd ratio, Exp(B), Z = Wald test statistics.

For variable categories, see Table 3. For variable categories, see Table 4. The reference category is the category indicating the lowest presence or absence of the characteristic.

**Table 6. Intervention features.**

| | | Non-CSP (n = 153) | | | | CSP (n = 83) | | | | Subgroup differences | |
| | | **n** | **% / median** | **95% CI / IQR** | **Intervention differences** | **n** | **% / median** | **95% CI / IQR** | **Intervention differences** | | |
| | | | | | **Test statistics** | | | | **Test statistics** | **Test statistic** | **p value** |
| | | | | | **p value** | | | | **p value** | | |
| Obtained intervention following randomisation % | Brief-PsT | 88 | 57.5 | | | 34 | 41.0 | | | $\chi^2_{(1)} = 5.926$ | p = .02 |
| | Short-PsT | 65 | 42.5 | | | 49 | 59.0 | | | | |
| Psychotherapy sessions, # *median* | Brief-PsT | | 4 | (2–6) | U = 1819.0 | | 5 | (3–6) | U = 541.0 | U = 1170.0 | p = .06 |
| | Short-PsT | | 6 | (3–10) | p < .001 | | 8 | (4–13) | p = .007 | U = 1342.5 | p = .15 |
| Time span, weeks # *median* | Brief-PsT | | 13.6 | (7.5–23.8) | U = 1731.0 | | 17.2 | (8.5–34.3) | U = 572.5 | U = 1227.0 | p = .12 |
| | Short-PsT | | 22.4 | (14.2–45.6) | p = < .001 | | 34.7 | (16.4–60.1) | p = .01 | U = 1349.0 | p = .16 |
| Optional physiotherapy consultation, % | Brief-PsT | 2 | 2.3 | (0.3–8.0) | $\chi^2_{(1)} = 0.12$ | 6 | 17.6 | (6.8–34.5) | $\chi^2_{(1)} = 0.43$ | $\chi^2_{(1)} = 7.12$ | p = .008 |
| | Short-PsT | 3 | 4.6 | (1.0–12.9) | p = .73 | 5 | 10.2 | (3.4–22.2) | p = .51 | $\chi^2_{(1)} = 0.62$ | p = .43 |
| Group education, participation % | Brief-PsT | 80 | 90.9 | (82.9–96.0) | $\chi^2_{(1)} = 0.55$ | 33 | 97.1 | (84.7–99.9) | $\chi^2_{(1)} = 0.02$ | $\chi^2_{(1)} = 0.61$ | p = .44 |
| | Short-PsT | 62 | 95.4 | (87.1–99.0) | p = .46 | 46 | 93.9% | (83.1–98.7) | p = .89 | $\chi^2_{(1)} = 0.00$ | p = 1.00 |

Intervention group and subgroup differences of # psychotherapy sessions and time span were evaluated with Mann–Whitney U test.

Differences in obtained intervention and participation rates are based on Chi-squared test with Yates' continuity correction.

### Comorbid spinal pain as a treatment effect modifier

Table 6 shows some intervention features. Significantly more non-CSP than CSP participants were allocated to Brief-PsT. In the non-CSP, the participants in the Brief-PsT group were slightly younger (38.3 vs 42.3 year, *p = .02*), but not in the CSP group (43.8 vs 43.5 years, *p = .45*).

Within both intervention groups, a non-significant trend of more psychotherapy sessions obtained by the CSP participants could be observed. The same trend of more treatment in the CSP subgroup was observable in total treatment time-span. Intervention differences of number of psychotherapy sessions and treatment duration were, as intended, significant in both subgroups. Nearly all participants had participated in the transdiagnostic group education. The participants were offered an optional physiotherapy consultations as well, only a minority requested this.

Positive treatment response rates are shown for the two intervention arms within both subgroups in Table 7. Time effects of WP from baseline to two-year follow-up were significant in all groups (Non-CSP: *Z = -6.2, p < .001* and *Z = -6.1, p < .001*; for CSP: *Z = -3.3*, p = .001 and *Z = -3.3, p < .001*) both arms, for both subgroups. Within both subgroups, no differences in WP improvements were found between the two intervention arms. Although CR2yr was higher in Brief-PsT in both subgroups, the differences were not significant at subgroup levels. Non-CSP showed highest treatment response rates in all conditions, but was only significant in WP following Short-PsT. Size and direction of the differences in treatment responses were comparable; this was confirmed in the non-significant results of the logistic regression analyses in which treatment modality was included as an interaction term (see Table 5). CSP could not be regarded as a moderator of treatment response. Any presence of SP at two-year follow-up was not related to intervention group either; among non-CSP SP was 35% and 43% (*p = .60*) in Brief-PsT and Short-PsT respectively and in CSP 74% and 71% (*p = 1.00*).

## Discussion

To our knowledge, this is the first study analyzing the impact of comorbid spinal pain on outcomes of brief and short psychotherapy for depression. The baseline characteristics of

**Table 7. Full work participation and clinical remission rate at two-year follow-up, stratified for the presence of CSP and intervention arm.**

| | Non-CSP | | | | | | CSP | | | | | | Subgroup differences | | |
|---|---|---|---|---|---|---|---|---|---|---|---|---|---|---|---|
| | Total | Positive treatment response | | | Intervention group differences | | Total | Positive treatment response | | | Intervention group differences | | | | |
| | | n | % | 95% CI | OR | Test statistic | P | n | % | 95% CI | OR | Test statistic | p | OR | Test statistic | p |
| **Full work participation at two-year follow-up**[a] | **122** | | | | | | | **73** | | | | | | | | |
| Brief-PsT | | 79 | 88.6 | 79.5–94.7 | 1.29 | U = 2641 | .39 | 32 | 71.9 | 53.3–86.3 | 0.87 | U = 795 | .72 | 0.33 | U = 1204 | .08 |
| Short PsT | | 55 | 90.9 | 80.1–97.0 | | | | 42 | 69.1 | 52.9–82.4 | | | | 0.22 | U = 1092 | .003 |
| **Clinical remission at two-year follow-up**[b] | **85** | | | | | | | **51** | | | | | | | | |
| Brief-PsT | | 43 | 76.7 | 61.4–88.2 | 0.40 | $\chi^2 = 2.86$ | .09 | 23 | 52.2 | 30.6–73.2 | 0.62 | $\chi^2 = 0.80$ | .37 | 0.33 | $\chi^2 = 3.11$ | .08 |
| Short PsT | | 42 | 57.1 | 41.0–72.3 | | | | 28 | 35.7 | 18.6–53.5 | | | | 0.42 | $\chi^2 = 2.29$ | .13 |

Prevalences are based on partial ITT analyses

[a] For work participation, intervention group differences and subgroup differences, changes of WP-state from baseline to follow-up were evaluated with Mann—Whitney U test—ITT

[b] Statistics for subgroup and intervention group differences in clinical remission are based on Chi-squared test with Yates' continuity correction

participants with CSP showed more frequently values related to poor health and CSP participants showed, indeed, poorer intervention response in both work participation and clinical remission. However, after adjusting for relevant factors, CSP was not a significant negative prognostic variable, neither for work-participation nor for clinical remission. The occurrence of CSP did not identify qualitative differences in responses of brief and short psychotherapy either.

## Comorbid spinal pain, associated with characteristics that are related to poor health and recovery

Confounding is by definition a nonissue in RCTs, but in the comparison of intervention effects within subgroups of a RCT it is essential to assess possible confounding factors that are related to both the occurrence of spinal pain and the intervention outcome measures. According to a literature review of Bair [4], about two out of three patients with depression also experience pain. This is about the same prevalence as in the present study. That about four out of ten had experienced substantial spinal pain last month even within the non-CSP-group, testifies to the high prevalence of pain in anxiety and depression. The prevalence is also considerably higher than found in the general population, where 13% reported substantial musculoskeletal complaints which included spinal pain, but also headache, shoulder pain, pain in arms and legs [45].

Previously, both older age and being female have been reported as risk factors for the presence of comorbid pain [46, 47]. In the current study there were no differences regarding gender distribution, but the subgroup with CSP was indeed characterised by a slightly higher age. This seems to reflect ordinary prevalence of spinal pain rates in working populations topping at middle age [48, 49].

Severity of depression was similar in both subgroups, high-moderate on average. This differs from other studies that have reported more depression symptoms with the presence of pain complaints [4, 50]. The present finding is surprising as it has been suggested that BDI-II may overestimate the prevalence of depression in patients with chronic pain complaints. They may report more items that address somatic symptoms like loss of energy, fatigue and sleep

disturbances [34, 51–53]. In addition, several cognitive and affective items (e.g. agitation, pessimism, irritability and concentration difficulty) could obviously be attributed to their chronic pain. Morley [54], however, reported that patients with depression and pain exhibit lower scores on the cognitive and affective items and no difference on the items reflecting somatic and physical function compared to depressed patients without pain. To our surprise, we did not detect any significant differences on the sub dimensions. In both subgroups, the mean score of the somatic dimension was the highest, followed by the affective and cognitive dimension. Similar findings were done by Demyttenaere [55]; they did either not find qualitative differences in depressive symptomatology between depressed patients with or without pain.

Besides spinal pain, the CSP participants reported also more severe anxiety complaints. This is expected from previous research; comorbid anxiety is frequently seen in both depression and chronic spinal pain or other pain [16, 50, 56–61]. However, like BDI-II, given a somatic symptom overlap, BAI may overestimate the level of anxiety in spinal pain patients. BAI comprises several somatic symptoms (like numbness, wobbliness in legs and inability to relax) which could be addressed to spinal pain as well. In addition, increased level of anxiety scores in participants with spinal pain may be related to lack of (pain) control, health related rumination, worry concerning cause and consequences of their spinal pain rather than symptoms of episodic or generalized anxiety. This was not directly evaluated in this current study, but we have seen that CSP participants reported a similar degree of fear-avoidance beliefs and concern as the non-CSP participants did.

Yet, the CSP participants were, understandably, more dissatisfied with their physical health. They also attributed more symptoms to their health problem, evident in a higher identity score of BIPQ and the comparative higher presence of additional SHC's. In the literature, increased prevalence of other chronic somatic symptoms associated with the co-occurrence of pain and depression has been reported before [47, 50].

The participants with CSP had more pessimistic beliefs of the timeline of their health complaints. Though, in spite of more registered sick leave in the previous year they had still had the same expectations concerning expected duration of their sick-leave. Increased pessimistic timeline as well as long sick leave has been reported before in comorbid LBP and depression compared to participants experiencing just one of these health problems [47, 62]. Comparable sick-leave expectations was surprising, especially given the fact that they also experienced higher physical workload. In spite of this physical work load, their level of fear-avoidance beliefs related to work was comparable as well, and they reported higher job satisfaction. The latter may have turned their sick-leave expectations more positive.

## Impact on clinical remission and work participation

In the current study we analysed whether the presence of CSP at baseline modified long-term intervention response irrespective of intervention assignment. The hypothesis that CSP in depressed participants is associated with prolonged sick-leave, was confirmed. Two years after baseline the probability to be on sick-leave is more than doubled for participants with CSP compared to the participants without CSP. However this was rather explained on the association with age; the CSP participants were four years older on average. Only age showed significance after inclusion of relevant covariates. Our expectation that CSP is an important factor for impeding long term clinical remission of mental health problems could not be confirmed either. It was, indeed, more likely for CSP participants to still report values of depression and/ or anxiety at a clinical level at two-year follow-up, but this was explained by higher baseline anxiety among participants with CSP. The absence of CSP as a prognostic factor for reduced remission in the multivariate analysis seems conflicting to general findings in depressed

patients with comorbid chronic pain [4, 15, 63–68]. Though, as far as we know, baseline anxiety is usually not included as a covariate in examining the effect of co-occurrence of spinal pain and depression for clinical remission.

Previous research on, prognostic factors of WP and clinical remission in depressed participants with CSP has been limited, so in this discussion we mainly compare results with evaluations in populations with either depression or spinal pain. The deviating characteristics of the CSP subgroup mentioned earlier are generally negatively associated with WP [2, 69–72], clinical remission of depression [15, 65, 73–78], musculoskeletal pain or disability [79–82], in cohorts with single or both complaints [83, 84]. That presence of CSF more than double the risk of sick leave in a clinical sample with high-moderate psychiatric symptom load testifies to the importance of developing treatments for combined symptomatology. This is comparable to the increased risks for severe disability reported by Scott [82]. Compared to the general population, the risk in persons with mental health disorders alone was four times higher, more than three times higher risk in spinal pain, but nine times as high risk when both conditions were combined.

The lack of significance of the various potential confounders we found in step 1 with the outcome measures surprised us. Still, several clinical, health management and work factors were associated with both presence of CSP and with clinical remission. However, as mentioned, the multivariate analyses revealed baseline anxiety as the only significant negative prognostic factor for clinical remission. The significance of anxiety for CR2yr in CSP is also revealed in the portion of participants with clinical anxiety scores at follow-up. Even though the number of participants with anxiety was lower in the non-CSP group than the CSP group at baseline, the reduction in prevalence was larger in the former group. In the CSP group half of the participants still suffered from anxiety. For depression, where the inclusion criteria determined a 100% prevalence at baseline, almost three quarter of the patients in the non-CSP remitted, while it was just over half in CSP.

The present findings of higher baseline anxiety in CSP, rendering the impact of CSP on symptom remission as insignificant, is complementary to previous research reporting negative prognostic value of baseline anxiety on pain outcome [76, 83, 85]. The causative mechanism behind the association is not clear and may include several pathways. Reviewing this, Gureje [60] claims that the pattern of association supports a causal pathway that proceeds from mental disorder to chronic pain rather than the reverse. In a longitudinal study among individual with chronic pain it was found, indeed, that neither pain nor pain-related disability predicted depression/anxiety [86]. Conversely, symptoms of both depression and anxiety, prospectively predicted levels of pain and pain related disability. However, findings from a large longitudinal twin study [87] found that pain predisposed for depression or anxiety symptoms, indicating a relationship explained by shared pathophysiological pathways. Possibly, the co-existence of anxiety and pain may point to a reciprocal relationship in which the direction might be dependent on context. Negative health cognitive bias, shared neurobiological features and somatization processes may mediate this relationship [86]. Based on the results of our present study, we will limit ourselves to informing clinicians that pain and anxiety may covary and that both would have to be addressed in therapy.

The findings of an association to presence of comorbidity in depressed participants with reduced WP is in line with previous research [2, 55, 88–90]. In the multivariate analyses, older age, satisfaction with physical health, and previous sick leave were the only covariates worth considering in our study. Significance of prior sick leave for WP has been reported earlier, both in populations with mental health disorders and with spinal pain [2, 69, 72, 91, 92], but was not confirmed here. Previous research has indicated an association of both the presence of work related fear-avoidance beliefs and self-reported high physical work load with previous

and future longer sick leave [93, 94]. In the current study, however, physical workload was not associated with WP at follow-up and fear-avoidance beliefs did not deviate between subgroups and can by definition not confound the relation between CFS and WP and were therefore not further considered as relevant as covariates in this current study.

Likewise, it is unsure whether improvements in pain and anxiety helps the patients' depressive symptoms or vice versa, or whether a common other factor is related to the severity and response of both depression, anxiety and pain [15]. Insomnia or fear-avoidance beliefs might be such a factor; related to both reduced mental health and chronic pain and reduced treatment effectiveness [95]. However, both factors were reported equally high in both subgroups and therefore not included as potential covariates.

Besides the mental health complaints, sleep quality and fear-avoidance beliefs, it could have been interesting to follow changes of SP, health management factors and WP during the intervention and follow-up period since they all have the potential to mediate intervention response. Longitudinal evaluations have shown that a reduction in depressive symptoms usually is associated with improved work participation [2, 96]. The synchronicity of change, however, might be delayed [88, 97] by other factors that influence work participation, like symptom acceptance, work load and self-efficacy [98]. Notable therefore was that the proportion of participants with full WP was higher compared to the proportion that was clinically recovered at follow-up in our study. This applied to both subgroups.

Another surprising and intriguing finding was that in non-CSP, full WP and CR were uncorrelated; the majority (91%) was at work, regardless of the presence of any depression, anxiety or spinal pain symptoms. In CSP, a similar proportion (89%) of the participants that had achieved clinical remission of mental health symptoms was at work as well. In this subgroup, however, the presence of depression and/or anxiety was associated with WP. The reduced WP in the subgroup CSP seems to be attributable to the participants that had not achieved remission at follow-up; only half realized WP. Logically, one would explain this by the presence of substantial spinal pain which was not part of the clinical remission status. However, the spinal pain showed only a tendency to be weakly related to full WP in CSP. The low number of participants that could be included for the analyses might be a reason for not detecting a significant association here. The lack of association between WP2yr and CR2yr in non-CSP is interesting. Both interventions seemed effective in the non-CSP participants for acquiring acceptance that subjective health complaints are normal features of human life and consequential assurance that continuing normal life and being at work may be crucial in managing these. In the CSP participants this seemed less successful; perhaps the additional burden of having pain issues are more difficult to handle at work, especially in light of the higher reported physical work load.

## Implications for psychotherapy

Although rates of comorbidity are usually high in RCTs with depressive participants, focus on comorbidity as moderator of treatment outcome has been scarce in the literature. Two systematic reviews indicated that depression and anxiety can be treated efficaciously with brief psychotherapy [22, 23], however possible impact of comorbidities were not discussed and the therapy was compared to less intensive approaches only.

In this pragmatic randomized trial, Brief-PsT had earlier shown its superiority for the whole group in both short term WP and long term CR [27]. Pain comorbidity generally seems to induce reduced outcome and an increased number of sessions [4, 99]. Therefore, we wanted to test out the possibility that behind the general superiority of brief psychotherapy shown in our previous study, nevertheless was hidden an opposite effect for those with comorbid CSP.

This expectation was motivated by the general rule in medicine that there is a relationship between severity of illness and treatment dose.

In the Short-PsT, with more extended focus, there was besides coping of mental health and challenges concerning WP, emphasis on an extensive anamnesis and possibility to focus on previous or current challenging psychological issues related to their current problems. This might typically apply to the impact of their pain complaints on work and life situation. However, the hypothesis of the moderating impact of CSP on psychotherapy type effectivity could not be confirmed in the present study. Similar to the whole group, within both subgroups, WP2yr did not differ between the two psychotherapy interventions either. On the other hand, within both interventions a comparable non-significant trend of reduced CR2yr in CSP was found, with a trend of greater clinical effectiveness of Brief-PsT in both subgroups. The latter was found in the primary study as well.

Decisions of therapy termination were done cooperatively by the psychotherapist and participant, usually because of remission or lack of improvement. In both sub-groups, the number of sessions in Short-PsT were considerably lower than aimed at. Although Brief-PsT was a management directive and standard procedure in our clinic, the psychotherapists had aspired to be allowed to expand both focus and number of sessions to improve outcome. Therefore, this substantial reduced applied number of sessions in Short-PsT was unexpected, especially in the CSP subgroup. We are unsure whether spinal pain and related problems had been discussed in particular in the psychotherapy, but there had certainly been the possibility to apply more sessions and address this more thoroughly in the psychotherapy.

Our outpatient clinic is a collaborative initiative of the psychiatric and physical medicine department, treating patients with mental health and/or musculoskeletal complaints. Nevertheless, it seemed not sufficiently effective for the depressed participants with comorbid spinal pain. The transdiagnostic group education prior to the psychotherapy, had a narrow focus on normalisation of common both mental health and pain complaints. The participants were supplied with updated evidence-based knowledge concerning depression, anxiety, musculoskeletal pain, and various approaches to manage common health challenges [27]. In addition, dependent on personal needs and interests, the participants had been offered an optional individual physiotherapist consult. They could participate on their own initiative or on their psychotherapist's recommendation. This consult consisted of a clinical examination with educational approach aimed explicitly at supplying the participants with insight and understanding of their spinal pain condition in order to remove potential uncertainty and fear and induce acceptance and coping with their pain. Acceptance of pain is after all seen as an independent prognostic factor of mental well-being, beyond the impact of pain severity [100]. Possibly, the majority of the CPS participants considered their insight as sufficient to accept and manage their spinal pain after participation in the group education; only a minority of the CPS participants had applied for an additional physiotherapist consult.

Still, it might have been possible that during the psychotherapy, their spinal pain and related concerns had not been sufficiently addressed, reassured and linked to their mental health symptoms with an effective explanation. Consequently, this may also have prolonged duration of the depressive or anxiety episode. The necessity of incorporating of assessment and treatment of both depression, anxiety and pain has previously been mentioned in the literature by others as well [4, 7, 15, 61, 66, 82, 99–101]. But, although cognitive behavioral therapy (CBT) often is recommended in long-term spinal pain [102–105], evidence of long-term benefits has been lacking [106–109]. The CBT may focus on helping the patients to change their interpretation of the pain and associated fear, symptom focusing, and avoidance [110, 111]. Adding CBT to a brief intervention comparable to the physiotherapy consult in this current study, or to

group physical exercise or other routine physiotherapy did not have any additional long-term effect on pain, depression, functioning and work participation in chronic LBP [110, 112, 113].

In the future, for the patients with spinal pain we may consider to put more emphasize on the value of a thorough reassuring clinical examination in addition to the referral for psychotherapy. Specific further cognitive behavioral focus on spinal pain and its consequences seems of little benefit. Still, the psychotherapists might be more aware of potential presence of comorbid health problems, particularly persistent pain and insomnia, address these and includes them in a normalization approach of common mental and physical health symptoms.

To explain the lack of convincing effectivity of Short-PsT with respect to clinical remission, we suggested earlier that the additional focus on past and current problematic issues in Short-PsT may, to a certain extent, rather have led to perpetuated worrying, ruminating and focusing on symptoms [27]. It has been suggested that more intensive interventions for depression may result in larger treatment benefits of comorbid symptoms, including pain [61, 114], but among the participants of this study, reduced treatment response in participants with substantial spinal pain was not related to type of intervention either.

In this pragmatic RCT, over a third of the participants that visited our outpatient clinic primarily because of depression, reported concurrent long-lasting spinal pain. That they recovered less both with reference to work-participation and clinical remittance regardless of therapy type, lead us to speculate what could be improved in both treatments. Exposure therapy is common in anxiety treatment, but additional exposure to pain eliciting situations has not been part of any of the present therapy models and obvious seems to be a plausible way to improve effectivity. However, effectivity in chronic nonspecific LBP has generally been modest and of short duration [115]. Graded exposure therapy in spinal pain focuses on the activities that expose the patient's fear. In a recent trial CBT and exposure therapy were compared in patients with high fear-avoidance levels. Exposure and CBT did not differ in reduction of pain intensity or disability and Exposure was less effective in enhancing coping strategies [116]. In this current study, however, both subgroups reported modest FABQ scores that were not related to anxiety severity. So we consider it as doubtful whether exposure therapy may be effective in our CSP subgroup. Still, it is of importance of explicitly targeting spinal pain problems and their potential hindrance for work participation and clinical remission in further research to develop effective treatment alternatives for depressed patients with CSP.

## Limitations

The participants in this study were primarily referred to the transdiagnostic outpatient clinic because of their mental health complaints. Therefore, spinal pain was defined as a comorbid complaint to depression. This may not have been correct as neither the chronologic aspects of the two conditions nor etiological associations have been evaluated. It might have been more correct to refer to the presence of spinal pain here as a multimorbid condition or label spinal pain as a co-occurent complaint without any implicit ordering [117].

Several studies suggest that after achieving recovery from depressive symptoms, impaired work functioning may still persist [118, 119]. In the current evaluations, a positive WP response was operationalised as full-WP only. However, from the main study we know that only a minority (<5%) had partial WP at two-year follow-up [27]. Although we did not assess whether this differed for the CSP vs. non-CSP, we assume that this did not have a significant impact on the presented results. Actual work functioning and any change of employment were not evaluated either.

Although subgrouping was based on a pre-randomized baseline characteristic, a point of concern is that the decision to perform subgroup analyses was post hoc. The RCT was

therefore possibly not adequately powered to do so. Consequently, the probability of generating 'false-negative' results in the analyses in this current study are substantial.

Because the subgrouping of CSP was not counted for by including it as a stratification variable in the randomization, distribution of the CSP participations was not balanced among the two intervention arms. Another consequence of post hoc subgrouping is that CSP could not be operationalized exactly. The participants that reported substantial spinal pain during the preceding months and had a current musculoskeletal pain episode of at least three months at baseline assessment, were labeled as CSP. This may involve the possibility that participants with spinal pain only during the previous month and with other musculoskeletal pain the previous months were falsely identified as having CSP. However, this is not so likely; disabling spinal pain is often non-specific and more frequently accompanied by other musculoskeletal pain than alone standing [120, 121]. Hence, it may be likely that several of the CSP participants were suffering from other musculoskeletal pain as well. Although sensitivity analysis for subgroup are especially relevant for post-randomization groups, we did however, perform sensitivity analyses to see how redefining the threshold for subgroup defining changed the observed subgroup and intervention outcome differences. For all four variants, contrasts and level of significance between subgroup outcome was greatest in the subgrouping that is reported here.

A strength, however, is that we have included covariates in the logistic regression analyses of treatment responses in the subgroups that revealed other significant characteristics that were related to CSP and worse outcome. Although the prognostic and effect modification analyses were not explicitly planned when designing the RCT, the choice of outcome measures seemed comprehensive enough to suggest as possible covariates according to current theory and experience. In the logistic regression models, only the variables which were significant for both the presence of CSP and the dependent outcome variables for the whole group were included as covariates. We might herewith have overlooked potential covariates which were equally present in both subgroups, but of which their impact on intervention effect was mediated by the presence of CSP.

Although randomization had been applied to balance the expected distribution of the baseline characteristics, patients in the Brief-PsT in the primary study, were slightly younger. This was also the case for non-CSP in the current study. This may have affected WP2yr, as we can see from the analysis of its association with treatment response. The participants with full WP were slightly younger than the participants on sick leave.

The modest response rates of the questionnaires at two-year follow-up, that included BDI-II and BAI measures which generated the clinical remission rate is an important limitation worth considering. Essential is here that losses to follow-up were neither related to the presence of comorbid CSP, nor to treatment assignment, to other clinical baseline measures or to WP response; the reasons for loss to follow-up could be considered as 'missing completely at random'. Although the observed differences are of clinical significance, the relatively small sample size of the clinical response measure, has restricted the power to detect possible relevant additional covariates and the role of CSP as a significant prognostic factor or treatment effect modifier moderator. The results of this study may be regarded as explorative and hypothesis generating.

## Conclusion

The literature regarding tight associations between anxiety, depression and pain is overwhelming, telling us that mental health affects pain and pain affects mental health. Although this calls for psychotherapy as possible treatment, the present study confirms previous findings that subjects with spinal pain, anxiety and depression profit less from psychotherapy than subjects

with anxiety and depression alone. Presence of CSP results in less work participation two years after the therapy and worse remission of mental health symptoms as well. The association of older age and of more severe baseline anxiety are of respective relevance for this reduced effectiveness. The expectation that more severe problems demand more extensive therapy, i.e. that subjects with comorbid CSP would profit more from a lengthier therapy process was not confirmed. Both the brief and the short-therapy format was equally less effective in treating comorbid CSP. As the length and focus of therapy, i.e. either acceptance and normalizing or additionally addressing psychologically central themes, did not mediate effectivity in treating comorbid CSP. Improvement of therapy for the CSP group must probably be sought by trying out other techniques.

## Acknowledgments

We would like to thank the participants for taking part in the study, and all therapists involved for their inspiring and enthusiastic help and participation. Also, thanks to Eivind Andersen for initiating the RCT and stimulating cooperation.

## Author Contributions

**Conceptualization:** Marjon E. A. Wormgoor, Aage Indahl, Jens Egeland.

**Data curation:** Marjon E. A. Wormgoor.

**Formal analysis:** Marjon E. A. Wormgoor.

**Methodology:** Marjon E. A. Wormgoor, Aage Indahl, Jens Egeland.

**Project administration:** Aage Indahl.

**Supervision:** Aage Indahl, Jens Egeland.

**Writing – original draft:** Marjon E. A. Wormgoor.

**Writing – review & editing:** Marjon E. A. Wormgoor, Aage Indahl, Jens Egeland.

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
