## [Decision Letter · Decision Letter 0]

15 Sep 2021

PONE-D-20-20567The impact of comorbid spinal pain in depression on work-participation and clinical recovery following brief or short psychotherapy. Secondary analysis of a randomized controlled trial with 2-year follow-upPLOS ONE

Dear Dr. Wormgoor,

Thank you for submitting your manuscript to PLOS ONE. After careful consideration, we feel that it has merit but does not fully meet PLOS ONE’s publication criteria as it currently stands. Therefore, we invite you to submit a revised version of the manuscript that addresses the points raised during the review process.

The manuscript has been evaluated by three reviewers, and their comments are available below.

The reviewers have raised a number of major concerns. Specifically, they feel the manuscript requires significant work in improving the flow, presentation, and clarity of the manuscript. In addition, they note the need for more specific detail in the methodology and statistical analyses. Reviewer #3 in particular raises a number of significant points that will need to be addressed thoroughly.  

Could you please carefully revise the manuscript to address all comments raised?

We look forward to receiving your revised manuscript.

Kind regards,

Avanti Dey, PhD

Staff Editor

PLOS ONE

Journal Requirements:

Reviewers' comments:

Reviewer's Responses to Questions

**Comments to the Author**

1. Is the manuscript technically sound, and do the data support the conclusions?

Reviewer #1: Partly

Reviewer #2: Partly

Reviewer #3: Partly

2. Has the statistical analysis been performed appropriately and rigorously? 

Reviewer #1: No

Reviewer #2: I Don't Know

Reviewer #3: No

3. Have the authors made all data underlying the findings in their manuscript fully available?

Reviewer #1: Yes

Reviewer #2: No

Reviewer #3: Yes

4. Is the manuscript presented in an intelligible fashion and written in standard English?

Reviewer #1: Yes

Reviewer #2: No

Reviewer #3: Yes

5. Review Comments to the Author

Reviewer #1: The manuscript entitled ‘The impact of comorbid spinal pain in depression on work-participation and clinical recovery following brief or short psychotherapy. Secondary analysis of a randomized controlled trial with 2-year follow-up’ with the aim to assess the predicting and moderating impact of baseline comorbid long-lasting spinal pain (CSP) on long-term work-participation (WP) and mental clinical recovery (CR) following psychotherapy for depression.

The manuscript can be further improved in terms of presentation, flow and clarity.

Comments

Introduction

Page 5 Line 79-83, the sentences require revision.

Methods-Potential covariates

Page 8 Line 167, biPQ to be written as BIPQ throughout the manuscript.

Page 9 Line 189 & Line 191-192, the sentence not clear and requires revision.

Statistical analyses

Page 10 Line 201-202, for Pearson’s Chi square tests, Student t-tests and Mann–Whitney U tests’ the words tests to be replaced with test.

Page 10 Line 202-203, based on CONSORT guidelines, statistical test for group comparison at baseline to be avoided.

Page 10 Line 208, for continuity correction, the word Yates’ to be added.

Page 10 Line 214, Chi-squared test with Yates’ continuity correction.

Results

At least one decimal point for percentages to be provided and to be applied throughout the manuscript.

Decimal points for p values to be standardized.

Page 12 Line 252, reasons lost to follow up to be stated.

Page 12 Line 257-258, what the figures represent i.e median to be clearly stated.

Page 13 Table 1, what range/coding/range 19-64 and difference refers to, to be clearly denoted. Median/mean with or without italicized and decimal points for p values to be standardized. Technically p value cannot be zero (to use symbol < - see also Line 302). The missing value to be denoted.

A description of information on missing data to be provided such as percentages of missing data, pattern etc

Page 12 & 13 Line 265-274, Page 14 & 15 Line 292-297, Page 15 Line 299-309, Page 19 Line 354-364, results to be presented in table form.

Page 14 Line 291-292, based on the tables sequence, results for Table 2 to the presented first.

Page 15 Line 301-302, sentence requires revision.

Page 15 Line 306-307, the figures to follow the figures in table.

Page 16 Line 324, Table 1 * to be written as Table 1 p values denoted with symbol *

Page 17 Line 333, phi to be replaced with symbol φ

Page 19 Line 369, table to be cited.

Page 15 & 16 Table 2, the table requires cosmetic changes. Spacing to be provided to separate the item variable. Font size to be standardized. Symbol =< to be replaced with symbol ≤ (likewise in the text). Test statistics and p values to be differentiated.

Page 16 Table 3, for positive treatment response, n to be stated apart from percentages. Test statistics and p values to be separated. Line 321, sentence incomplete. Comparison of what to be clearly stated.

Page 17 Table 4, under Yes, n to be stated apart from percentages. Statistical test to be denoted in the table footnote. Decimal points for p values to be standardized. Since LISAT and satisfaction with physical health involve mean, the statistical test for these two to be clearly denoted in the table footnote. For Last year sick leave, undo the italicized for ≥ 100 days. Likewise, for some cited references.

Page 18 & 19 Table 5, the presentation could be improved. The title logistic regression results too short and to be expanded. The categories for each variable to be presented and reference category to be highlighted or alternatively, the coding/reference category for the variables to be denoted in the table footnote. More information to be provided i.e SE, pseudo R squared, model fit, etc

Discussion

Page 20 Line 380, the word correction to be replaced.

Figure 1, n for subjects and n for independent and dependent variables to be clearly differentiated.

[ ] to be used for references cited in the text. table to replaced with Table.

Tables look congested and requires cosmetic changes in terms of presentation, spacing, font size etc,

Reviewer #2: The paper presents an interesting topic and is a comprehensive study of work participation and spinal pain and depression comorbidity. However, I have some concerns about the novelty of some of the results in this paper, and there are some aspects about the presentation of the paper, method/design and readability/writing that require further work and discussion. Some general comments and recommendations for each section of the manuscript are outlined below:

Abstract:

- For readability, it is preferrable that the number of abbreviations is reduced as much as possible. Also some of the abbreviations are not described (e.g., cOR)

- The design of the current paper is not well described in the abstract (nor in the following manuscript). You describe that it is a secondary analysis, however, it is not clear how you are analyzing the sample (i.e., if you are looking at the different interventions or the study sample as a cohort). See further comments about this in later sections.

Introduction:

- In general, the written language throughout this paper could be improved. Some of the content is unclear and it could be slightly more academic, more precise and shortened.

- Aim: The main aim about investigating the predicting and moderating impact of baseline presence of CSP on work participation and clinical recovery are interesting. However, when you later list up your hypotheses, these includes largely different questions, and I’m concerned about the readability of the study when you include all these hypotheses in one paper.

- Aim: Second, I’m concerned about the novelty of the first hypothesis, and suggest that this can be toned down in the present paper.

- Aim: third, all the different hypothesis demands to use different methods/design, and as previous mentioned, it is not clear whether you want to use the study sample as a cohort or as a subgroup analysis of the randomized trial.

Method:

- In general, a more detailed description of the study sample and method that you use for this paper are needed. The sampling procedure and randomization (if you want to analyze this as a randomized trial, though..) are lacking, it should not be necessary to look up the primary paper to evaluate this. Further, including both prediction analyses and looking at subgroup differences of the two treatment modalities seems a bit comprehensive. Is it possible to separate these aims into different papers? It would ease the readability and make it more clear.

- Potential covariates (line 145): You describe that you have selected apriori variables based on clinical experience and previous studies, but you don’t use all of them as confounding variables in the analysis. What is the rationale behind this? Selecting confounders based on statistical test is not recommended. I believe that some of the tests are used for selecting variables in the prediction model, however this is not clearly defined and explained.

- Statistical analyses: Not clear what you mean in line 198. In the paragraph from line 199 – 204, you describe several tests for comparison, however, you don’t describe what you are using this for. Is this to answer the first hypothesis? (if so, maybe not needed?) or is it part of the selection of variables for later analysis? This should be clearly described. In the second last paragraph (line 226-229) you describe the method of analyzing if the subgroups respond differently to the two interventions. I wonder if this should be analyzed the same way as the original RCT instead of using logistic regression? This is not my field of expertise, so further discussion with a statistician is suggested.

Results:

- The result section can be shortened. For instance, you mention the 30-days prevalence of neck, upper and lower back pain, which is not of major relevance to your aim (line 247).

- Not clear what you mean in line 255, and maybe not needed to include.

- Section about non-repliers can be shortened (line 257-262)

- Comparative characteristics according to presence of comorbid spinal pain: As previous mentioned, I question the novelty of these results. At best, I think it should be largely shortened.

- Table 1: This also refers to the question about you analyzing the sample as a cohort or as a randomized trial with subgroups. Since one of your research questions is to look at different subgroup effect of two interventions, it would be useful show the baseline characteristics stratified into the intervention groups as well as your subgroups.

- The order of the tables is not chronological, you refer to Table 3 before Table 2 in the text.

- The OR in text is not found in the tables (line 293)

- Table 3: the covariates/confounders in the analyses are not described in the table.

- Table 4: This table includes information that is not part of your aim (except for the first line: CSP and treatment response). As far as I know, you do not aim to look at gender, age, job satisfaction etc. and the association with work participation and clinical recovery. I suggest removing this or to justify and describe in you aim.

- Table 5: Lack a descriptive legend for the table. Are the analyses used here adjusted? This is not described.

- Line 365-366: I cannot find this results in any table.

Discussion:

- Comments about the novelty of some of the results also apply for the discussion section.

- Line 432-433. The wording should be rephrased, it is not correct to state that the probability of sick leave was three times as high, based on the results presented in this paper.

- Line 435: You present a new result in the discussion here? (aOR=2.7).

- Line 449-451: for clarity, you should describe for what outcome measure the OR refer to.

- Line 454-455: You write “…the multivariable analyses revealed baseline anxiety as the only significant predictor for clinical recovery”. But is it an aim in this study to identify possible prognostic variables?

- Limitations: You mention multiple limitation in this study, which is relevant, however, you don’t say anything about how these interventions influence the results and whether you have addressed any of the limitations (e.g., by performing sensitivity analyses). It would be relevant to perform sensitivity analyses regarding potential covariates, or to identify if participants suffering from widespread pain would influence the results.

- Nothing is mentioned about “breaking” the randomization by creating subgroups within the randomized controlled trial. This is an important aspect when you compare the effect of the interventions between the subgroups.

- Nothing is mentioned about mechanisms behind the possible association between spinal pain, mental disorders and recovery and work participation. This would be useful to get a greater insight into the complex nature of this patient group.

Reviewer #3: Please see attached review comments. Thank you for the opportunity to review this manuscript and to read more about your research. Your research represents a valuable and considerable effort to consider the impact of pain on psychotherapeutic outcomes over a longer-term follow up and I feel that this is important and valid for publication, though the manuscript needs significant further reworking.

6. PLOS authors have the option to publish the peer review history of their article (what does this mean?). If published, this will include your full peer review and any attached files.

Reviewer #1: No

Reviewer #2: No

Reviewer #3: No

---

## [Author Response · Author response to Decision Letter 0]

31 Jan 2022

All comments are responded on and described in the cover letter and uploaded as a separate file labeled 'Response to Reviewers'

In the letter we applied colours to indicate comments from the editor and reviewers, our responses and changes in the manuscript.

I will copy these in here as well, but it probably will be somewhat unclear in this format.

Reviewer #1:

The manuscript entitled ‘The impact of comorbid spinal pain in depression on work-participation and clinical recovery following brief or short psychotherapy. Secondary analysis of a randomized controlled trial with 2-year follow-up’ with the aim to assess the predicting and moderating impact of baseline comorbid long-lasting spinal pain (CSP) on long-term work-participation (WP) and mental clinical recovery (CR) following psychotherapy for depression.

The manuscript can be further improved in terms of presentation, flow and clarity.

Comments

Introduction

Page 5 Line 79-83, the sentences require revision.

We have rewritten this section:

“The presence of CSP at baseline was hypothesized as a predictor of treatment response and as a moderator of response of brief and short psychotherapy. The following sub-hypotheses were examined: First, CSP is associated with prolonged sick leave and second, CSP hampers long-term clinical remission of mental health problems. Third, for the participants with CSP complaints, psychotherapy with a more extended focus is more beneficial considering both WP and clinical remission.”

Methods-Potential covariates

Page 8 Line 167, biPQ to be written as BIPQ throughout the manuscript.

Is corrected.

Page 9 Line 189 & Line 191-192, the sentence not clear and requires revision.

These sentences are rewritten: 

“Workload was evaluated with the questions “My job is physically demanding” (#5, Effort Reward Imbalance questionnaire[1]) and “My job is mentally demanding”, dichotomized as ‘does not apply’, or ‘does apply’. Job satisfaction was assessed with the question “do you enjoy being at work?” Workplace uncertainty was evaluated with question #59 of General Nordic Questionnaire for Psychological and Social Factors at Work (QPSNordic) [2] “Are there rumours concerning changes at your workplace?”

Statistical analyses

Page 10 Line 201-202, for Pearson’s Chi square tests, Student t-tests and Mann–Whitney U tests’ the words tests to be replaced with test.

Is corrected.

Page 10 Line 202-203, based on CONSORT guidelines, statistical test for group comparison at baseline to be avoided.

We are aware of this, but we think that this applies for randomized groups in which observed baseline differences may occur by chance. However, in the manuscript we compare two subgroups that were created based on the presence of spinal pain. This makes it relevant to report statistical testing for group comparison at baseline here.

Page 10 Line 208, for continuity correction, the word Yates’ to be added.

Page 10 Line 214, Chi-squared test with Yates’ continuity correction.

Is corrected.

Results

At least one decimal point for percentages to be provided and to be applied throughout the manuscript.

Is corrected.

Decimal points for p values to be standardized.

We have standardized the decimal points as follows: P-values >.01 are expressed with 2 decimals, otherwise with 3-decimals except for p-values .044-.049. P-values less than .001, are reported as P<.001.

Page 12 Line 252, reasons lost to follow up to be stated.

In the flow diagram (fig. 1) the number of participants and reasons for lost to follow up were reported as “exclusion from analysis: “WP not available if moved out of the county, were NAV-employees or close family (n=11 and 9), CR not available; non-return questionnaire (n=55 and 45)”. We have not repeated this information in text.

Page 12 Line 257-258, what the figures represent i.e median to be clearly stated.

Is corrected.

Page 13 Table 1, what range/coding/range 19-64 and difference refers to, to be clearly denoted. Median/mean with or without italicized and decimal points for p values to be standardized. Technically p value cannot be zero (to use symbol < - see also Line 302). The missing value to be denoted.

We have changed the heading of the second colon in table 1 to “Range of values”. Further, we have standardized the type of measures and p values. In the third column, we have replaced the column with number “completed” with “Missing data n/n”, for each variable, the number of responders with missing values are reported, if deviating from zero

A description of information on missing data to be provided such as percentages of missing data, pattern etc.

As mentioned earlier, the reasons for missing questionnaires are given in the flow diagram. We have not any more detailed information concerning reason of further missing values of the individual items, but these seems small. However, in the second paragraph of the results section we describe differences in the responders and non-responders (=lost to follow-up=missing data) in detail. In addition, we have added to the section ‘Comparative characteristics according to the presence of comorbid spinal pain’:

“Missing values were, apart from the work characteristics items, close to zero percent and completely at random (Little’s test, p= .24).”

Results to be presented in table form:

It is always a dilemma how much information should be given in a table, and when to instead give additional information in text. We have aspired and attempted to make the tables as clear and structured as possible.

Page 12 & 13 Line 265-274

The results of 265-270 are presented in table 1, we now only report individual SHC ‘pain items’ items in the text.

Page 14 & 15 Line 292-297

Some statistic information are placed in a new table 2, (without intervention group information). The other results for interrelations between and WP2yr and CR2yr are hard to include in the table, but we think it is still interesting to present.

Page 15 Line 299-309

We have restructured and simplified Table 3 (former Table 2) as these lines refers to. Herewith we could include some of the textual information in the text. We have skipped the correlation data from the table and text.

Page 19 Line 354-364, 

A new table 6 describing intervention features is added. Numeric results and statistics are removed from the text.

Page 14 Line 291-292, based on the tables sequence, results for Table 2 to the presented first.

Table 2 is new and the rest of the information that was placed in table 3 is table 7 now.

Page 15 Line 301-302, sentence requires revision.

Is corrected.

Page 15 Line 306-307, the figures to follow the figures in table.

We rephrased this line, but the figures are not presented in the tables: 

“SP at follow-up did not affect WP2yr (non-CSP 90.9% vs. 91.3%, p=1.0; CSP 90.6 vs. 60.6% p=.14). But, among the participants with follow-up spinal pain, WP2yr was significantly lower in CSP (p=.01). Among the participants with follow-up spinal pain, WP was 91% in non-CSP and 61% in CSP (p=.012).”

Page 16 Line 324, Table 1 * to be written as Table 1 p values denoted with symbol *

Is corrected.

Page 17 Line 333, phi to be replaced with symbol φ

Is corrected.

Page 19 Line 369, table to be cited.

Referred to table 7 at start of the paragraph and in the previous line.

Page 15 & 16 Table 2 (now 7), the table requires cosmetic changes. Spacing to be provided to separate the item variable. Font size to be standardized. Symbol =< to be replaced with symbol ≤ (likewise in the text). Test statistics and p values to be differentiated.

We have tried to increase spacing between the variables and standardized the font size and italics. We replaced p=<.001 with p<.001.

Page 16 Table 3 (now 2 and 7), for positive treatment response, n to be stated apart from percentages. Test statistics and p values to be separated. Line 321, sentence incomplete. Comparison of what to be clearly stated.

Data of Table 3 are now in Table 2 and 7 and improved. 

Line 321 is adjusted.

Page 17 Table 4, under Yes, n to be stated apart from percentages. Statistical test to be denoted in the table footnote. Decimal points for p values to be standardized. Since LISAT and satisfaction with physical health involve mean, the statistical test for these two to be clearly denoted in the table footnote. For Last year sick leave, undo the italicized for ≥ 100 days. Likewise, for some cited references.

For each parameter we have indicated the total n (= number of available cases). We did not report the n in addition of the percentages as we think it would not improve the clarity of the table.

Statistical tests were described in the methods but added here as a footnote. LISAT (and illness perception) were based on mean scores, indeed, but dichotomized and analyzed likewise.

We are not sure what you mean with your last comment “Likewise, for some cited references”. 

Page 18 & 19 Table 5, the presentation could be improved. The title logistic regression results too short and to be expanded. The categories for each variable to be presented and reference category to be highlighted or alternatively, the coding/reference category for the variables to be denoted in the table footnote. More information to be provided i.e. SE, pseudo R squared, model fit, etc.

The title is expanded: 

“Table 5. Results of simple and multiple logistic regression analysis for the association between CSP and associated variables with the dependent variables WP2yr and CR2yr. In addition, the results of analysis of intervention interaction in the association of CSP with WP2yr and CR2yr.”

For each categorical variable included in table 5 only the presence of the specific characteristic (yes/no) is included, we did not include different categories. Therefore, in practice, the absence of the characteristic is the reference category. In the methods and table 4 we described the dichotomous split. We think it is too much to repeat this in this table, but in the footnote is now referred to table 4 for explanations of the coding/reference categories. We added:

“For variable categories, see table 4. The reference category is the category indicating the lowest presence or absence of the characteristic”

We have added Coef(B), SE and Nagelkerke R2

Discussion

Page 20 Line 380, the word correction to be replaced. 

“After correction for relevant factors, CSP was still a significant negative predictor for work-participation, but not for clinical remission”

We changed some variables in the logistic regression analysis. Now this sentence is not correct anymore and changed.

Figure 1, n for subjects and n for independent and dependent variables to be clearly differentiated

In figure 1, n reports the number of subjects included in the randomization process and the number of subjects with available data for analyses at the different moments and type of variables. To avoid confusion, we have removed the terms ‘independent’ and ‘dependent variables’ in the flow diagram and replaced it with ‘Baseline variables’ and ‘Two-year follow-up, outcome’, ‘work participation’ and ‘clinical remission’.

[ ] to be used for references cited in the text. table to replaced with Table.

Is corrected.

Tables look congested and requires cosmetic changes in terms of presentation, spacing, font size etc.,

We have tried to improve this. We expect that this will be further improved after possible production.  

Reviewer #2:

The paper presents an interesting topic and is a comprehensive study of work participation and spinal pain and depression comorbidity. However, I have some concerns about the novelty of some of the results in this paper, and there are some aspects about the presentation of the paper, method/design and readability/writing that require further work and discussion. Some general comments and recommendations for each section of the manuscript are outlined below:

Abstract:

- For readability, it is preferrable that the number of abbreviations is reduced as much as possible. Also some of the abbreviations are not described (e.g., cOR)

The number of abbreviations are reduced and statistics are deleted from the abstract as well as ‘cOR’.

- The design of the current paper is not well described in the abstract (nor in the following manuscript). You describe that it is a secondary analysis, however, it is not clear how you are analyzing the sample (i.e., if you are looking at the different interventions or the study sample as a cohort). See further comments about this in later sections.

The abstract and the methods are partly rewritten to clarify this.

In abstract: “Based on baseline assessment, the sample was subdivided into a subgroup with and subgroup without CSP. Work participation and clinical remission of depression and anxiety symptoms was assessed as treatment outcome at two-year follow-up. Across the intervention arms, simple and multiple logistic regression analyses were applied to evaluate predictive and moderating effects of CSP on the outcome.”

In statistical analyses: "Data of both intervention arms were pooled” ….. “Both intervention groups were combined in the analyses and regarded as one cohort.”

Introduction:

- In general, the written language throughout this paper could be improved. Some of the content is unclear and it could be slightly more academic, more precise and shortened.

We have worked through the manuscript and made many minor wording changes to shorten long sentences, omit modifiers and improve preciseness.

- Aim: The main aim about investigating the predicting and moderating impact of baseline presence of CSP on work participation and clinical recovery are interesting. However, when you later list up your hypotheses, these includes largely different questions, and I’m concerned about the readability of the study when you include all these hypotheses in one paper.

- Aim: Second, I’m concerned about the novelty of the first hypothesis, and suggest that this can be toned down in the present paper.

We have omitted the first hypothesis 

- Aim: third, all the different hypothesis demands to use different methods/design, and as previous mentioned, it is not clear whether you want to use the study sample as a cohort or as a subgroup analysis of the randomized trial.

I hope it is clarified better now in the revised method section. Basically, the study sample was used as a cohort, only for the analyses of CSP as a moderator response, type of psychotherapy was included in the analyses.

Method:

- In general, a more detailed description of the study sample and method that you use for this paper are needed. The sampling procedure and randomization (if you want to analyze this as a randomized trial, though..) are lacking, it should not be necessary to look up the primary paper to evaluate this. 

The study design, randomization and setting of the method section is partly rewritten to clarify this.

Further, including both prediction analyses and looking at subgroup differences of the two treatment modalities seems a bit comprehensive. Is it possible to separate these aims into different papers? It would ease the readability and make it more clear.

We do agree that the combination of both prediction analyses of general treatment effects and the analyses of subgroup differences of the two treatment modalities may be a bit comprehensive. However, we think these two focusses are very connected to each other and we prefer to present this in one paper. If the hypotheses that CSP is associated with prolonged sick leave and reduced long-term clinical remission of mental health problems are correct, it is logical to evaluate whether psychotherapy with a more extended focus is more beneficial.

- Potential covariates (line 145): You describe that you have selected apriori variables based on clinical experience and previous studies, but you don’t use all of them as confounding variables in the analysis. What is the rationale behind this? Selecting confounders based on statistical test is not recommended. I believe that some of the tests are used for selecting variables in the prediction model, however this is not clearly defined and explained.

In the new Statistics and Data analysis section, we have now added some more text regarding the selection of covariates. The text now reads: “Step 1 involved selection of potential covariates, based on the results of the association of baseline factors with both CSP, baseline WP and treatment response. To be comprehensive we chose a lenient level of significance for including possible covariates (p.15). In step 2, analyses (Chi-squared test with Yates’ continuity correction for all 2x2 analyses and Mann–Whitney U test for numerical variables) were applied to assess the associations with treatment response for both defined binary dependent outcome variables: WP2yr and CR2yr.

By emphasizing the lenient level of significance for qualifying for a candidate covariate, we want to signal that we also are concerned that we include all relevant covariates into the subsequent analyses. But, it is customary to have some selection procedure, usually a stricter significance level. By this procedure, we reduce the variables from 36 to 14 and to 11 finally possible covariates, which still is a large number. Therefore, the rationale was first to address all possible variables that could be relevant from review of what other researchers have looked at, but to show the reader why we do not follow up on those variables with low possibility in this material. Examples are “fear avoidance beliefs and “coworker support” which theoretically could meaningfully mediate return to work, but showed no difference between CSP and non-CSP and thus further analysis of this variable would merely add to the complexity (and not clarity) of the paper. 

- Statistical analyses:

Not clear what you mean in line 198. In the paragraph from line 199 – 204, you describe several tests for comparison, however, you don’t describe what you are using this for. Is this to answer the first hypothesis? (if so, maybe not needed?) or is it part of the selection of variables for later analysis? This should be clearly described.

As you suggested earlier, we have removed the first hypothesis. Descriptive statistics are still presented, to describe our population and the two different clusters in particular, and for selection of variables for later analysis. 

In the second last paragraph (line 226-229) you describe the method of analyzing if the subgroups respond differently to the two interventions. I wonder if this should be analyzed the same way as the original RCT instead of using logistic regression? This is not my field of expertise, so further discussion with a statistician is suggested.

We were interested to assess whether CSP moderates treatment effects as was hypnotized by expecting that brief psychotherapy is not sufficient (according to the conclusion of the primary RCT, brief psychotherapy was superior in treatment effect). Since treatment modality is included as an interaction term, we think that logistic regression is appropriate here.

Results:

- The result section can be shortened. For instance, you mention the 30-days prevalence of neck, upper and lower back pain, which is not of major relevance to your aim (line 247).

The 30-days prevalence of substantial neck, upper and lower back pain is part of the selection of the subgroup with CSP and we believe it is therefore appropriate to report this here. We have tried to include as much as possible in the tables and not to repeat that in detail in the text.

- Not clear what you mean in line 255, and maybe not needed to include.

This line presented gender differences in loss to follow up. We have rewritten this in the following line:

“In both subgroups, participants with available CR-2yr status were more often female (61.2 vs 45.5%, p=.09 and 62.2% vs. 44.2%, p=.04) and older (median 42.7 vs 36.5, p<.001 and 45.6 vs 40.5, p=.03) than the non-repliers.”

- Section about non-repliers can be shortened (line 257-262)

Unfortunately, we did not shorten this section as a compromise with one of the other reviewers who requested more information. We think it is informative.

- Comparative characteristics according to presence of comorbid spinal pain: As previous mentioned, I question the novelty of these results. At best, I think it should be largely shortened.

The section is shortened, we removed the data of the individual SHC items.

- Table 1: This also refers to the question about you analyzing the sample as a cohort or as a randomized trial with subgroups. Since one of your research questions is to look at different subgroup effect of two interventions, it would be useful show the baseline characteristics stratified into the intervention groups as well as your subgroups.

We consider it as most relevant to present baseline data and the subgroup differences. If we in addition should present the same table stratified for intervention arm we think this would be overwhelming. According to the CONSORT guidelines, it is not necessary to apply significance tests to show for comparison of randomised groups.

However, in the primary paper we found the participants in the Brief-PsT group were slightly younger. We found this for the non-CSP patients in this current paper as well. Apart from this, there had been no differences in any other demographic or clinical characteristics. We added in the results- ‘Comorbid spinal pain as a moderator of treatment response’ section: In the non-CSP, the participants in the Brief-PsT group were slightly younger (38.3 vs 42.3 year, p=.02), but not in the CSP group (43.8 vs 43.5 years, p=.45). 

In the Discussion –Limitations: “Although randomization had been applied to balance the expected distribution of the baseline characteristics, patients in the Brief-PsT in the primary study, were slightly younger. This was also the case for non-CSP in the current study. This may have affected WP2yr, as we can see from the analysis of its association with treatment response. The participants with full WP were slightly younger than the participants on sick leave.”

- The order of the tables is not chronological, you refer to Table 3 before Table 2 in the text.

This is changed

- The OR in text is not found in the tables (line 293)

The odds were calculated from the data of table 2. We replaced, however these odds and reported odds ratios (ORs) instead. We aimed at not repeating too much data from tables in the text and assumed it as appropriate to present these ORs only in the text.

- Table 3: the covariates/confounders in the analyses are not described in the table.

We are not sure what description you are aiming at. In Table 3 (no 2 and 7) are no covariates/confounders included in the analyses. All covariates selected for analyses in Table 4 has been described in the methods section and descriptive data was given in Table 1.

- Table 4: This table includes information that is not part of your aim (except for the first line: CSP and treatment response). As far as I know, you do not aim to look at gender, age, job satisfaction etc. and the association with work participation and clinical recovery. I suggest removing this or to justify and describe in you aim.

Gender, age, job satisfaction were in table 1 selected as possible potential covariates according to step 1 described in statistical analyses section (p<.15) and so part of the main objective to evaluate the predicting impact of baseline presence of CSP on treatment response.

- Table 5: Lack a descriptive legend for the table. Are the analyses used here adjusted? This is not described.

The title of Table 5 has been improved to clarify its contents: “Table 5 Results of simple and multivariate logistic regression analysis for the association between CSP and associated variables with the dependent variables WP2yr and CR2yr. In addition, the results of analysis of intervention interaction in the association of CSP with WP2yr and CR2yr.”

As a footnote it is added: “For variable categories, see Table 3. The reference category is the category indicating the (highest) presence of the characteristic.”

In the simple logistic regression analyses, the analyses were not adjusted, in the multiple logistic regression analyses they were adjusted. 

- Line 365-366: I cannot find this results in any table.

These results were not in table 4 as stated in the manuscript, but in table 2 (now 7). This is corrected. For clarification, the lines are rewritten.

“Table 2. shows WP and CR results for the two clusters within the two interventions. Within both clusters, no differences in WP improvements were found between the two intervention arms. However, WP improvement from baseline to follow-up was significantly lower in CSP compared to non-CSP.”

Discussion:

- Comments about the novelty of some of the results also apply for the discussion section.

We have improved the discussion

- Line 432-433. The wording should be rephrased, it is not correct to state that the probability of sick leave was three times as high, based on the results presented in this paper.

We think it was correct. The figures we referred to were 30% and 10% that had no full WP (= on sick leave). However, we have now included ITT results as well so we needed to moderate this probability (it is 2.3 for IIT analyses and 2.9 times for Partial ITT analyses). We now write:

“Two years after baseline the probability to be on sick-leave is more than doubled for participants with CSP compared to the participants without CSP ….”

- Line 435: You present a new result in the discussion here? (aOR=2.7).

aOR=2.7 (sick leave = not WP) is the inverse of OR 0.37 (WP) for CSP in the multiple logistic regression as was presented in table 5. However, this has now been removed from the discussion.

- Line 449-451: for clarity, you should describe for what outcome measure the OR refer to.

Following the suggestion by Reviewer 3, we have rewritten parts of the Discussion section in order to refer to fewer numbers, and focus on the substantial issues. The text at this point is rewritten and now reads:

“That presence of CSF more than double the risk of sick leave in a clinical sample with high-moderate psychiatric symptom load testifies to the importance of developing treatments for combined symptomatology. This is comparable to the increased risks for severe disability reported by Scott [82]. Compared to the general population, the risk in persons with mental health disorders alone was four times higher, more than three times higher risk in spinal pain, but nine times as high risk when both conditions were combined.” 

- Line 454-455: You write “…the multivariable analyses revealed baseline anxiety as the only significant predictor for clinical recovery”. But is it an aim in this study to identify possible prognostic variables?

No, it was not a direct aim to identify possible prognostic variables other than CSP. However, since this was a result of the multivariate analyses, we think it is relevant to report and discuss it.

- Limitations: You mention multiple limitation in this study, which is relevant, however, you don’t say anything about how these interventions influence the results and whether you have addressed any of the limitations (e.g., by performing sensitivity analyses). It would be relevant to perform sensitivity analyses regarding potential covariates, or to identify if participants suffering from widespread pain would influence the results.

We have performed sensitivity analyses to see how redefining the threshold for subgroup defining changes the observed subgroup and intervention for differences for WP2yr and CR2yr. 

In statistics section: “Sensitivity analyses were performed to see how redefining the threshold for subgroup defining changes the observed subgroup and intervention for differences for WP2yr and CR2yr. We repeated the analysis for four different thresholds: subgroups with the presence of only 1-month substantial spinal pain, of one-month any spinal pain, for one-month any spinal pain plus three months musculoskeletal pain or for only three months musculoskeletal pain.”

In results: “We have performed sensitivity analyses to check whether other subgrouping thresholds effect treatment responses. For all variants, contrasts and level of significance between subgroup outcome of WP2yr and CR2yr was greatest in the subgrouping that is reported here.”

In addition, we have performed some power analyses to check whether our sample sizes were sufficient to detect differences in WP2yr and CR2yr. Minimal sample size was 74 for WP2yr (our study 153 and 83) for WP2y, and minimal 66 for CR2yr (our study 85 and 51).

As expected, sample sizes in the trial arms are obvious to small (WP2yr: 79, 55, 32 and 42, CR2yr: 43, 42, 23 and 28)).

- Nothing is mentioned about “breaking” the randomization by creating subgroups within the randomized controlled trial. This is an important aspect when you compare the effect of the interventions between the subgroups.

We agree that the new division into non-CSV and CSV was performed post-hoc. Thus, we have added in the revised limitation section that:

“The results of this study may be regarded as explorative and hypothesis generating.”

- Nothing is mentioned about mechanisms behind the possible association between spinal pain, mental disorders and recovery and work participation. This would be useful to get a greater insight into the complex nature of this patient group.

In the Introduction, we have added a little more, mentioning the shared biological pathways and additive effects on symptoms, disability and impact on worse outcome:

“their shared biological pathways and symptoms has been pointed out frequently. It has been suggested that a bidirectional relationship exists between chronic spinal pain and depression, in which spinal pain is a risk factor for depression and depression is a risk factor for spinal pain. Previous research has suggested that comorbidity is associated with additive or synergistic effects of increased symptom severity and disability and a negative impact on clinical remission and quality of life outcomes following intervention.”

In the discussion, we have not directly discussed mechanism behind the possible associations between the complaints and treatment outcome. However, we had mentioned fear-avoidance beliefs as a relevant factor mentioned in the literature, but it seemed not of specific relevance in our study. Yet, we have now included insomnia as a potential factor as well, but this was not relevant either in our analyses either. Further, we included a short discussion around the relation of improvements in pain and anxiety:

“Likewise, it is unsure whether improvements in pain and anxiety improvement helps the patients’ depressive symptoms or vice versa, or whether a common other factor is related to the severity and response of both depression, anxiety and pain [15]. Insomnia or fear-avoidance beliefs might be such a factor; related to both reduced mental health and chronic pain and reduced treatment effectiveness [94]. However, both factors were reported equally high in both subgroups and therefore not included as potential covariates.

Besides the mental health complaints, sleep quality and fear-avoidance beliefs, it could have been interesting to follow changes of SP, health management factors and WP during the intervention and follow-up period since they all have the potential to mediate intervention response.”

We have also shortly discussed the possible mechanism concerning the relation of CSP and the observed increased baseline anxiety: “Taking these studies in consideration, the present findings of higher baseline anxiety in the CSP, rendering the prediction of CSP on symptom remission insignificant, is probably not a coincidence. An important question then is: why is anxiety seemingly specifically associated with CSP? Does CSP increase anxiety, or does anxiety cause spinal pain? It may depend on whom you are asking. Psychosomatic medicine would emphasize how anxiety give rise to musculoskeletal tension and pain [85]. Discussing this in a review, Gureje [59] states that the pattern of association supports a causal pathway that proceeds from mental disorder to chronic pain rather than the reverse. However, findings from a large longitudinal twin study showed a non-causal relationship between a history of chronic LBP and the future risk of depression or anxiety symptoms [86]. It was hypothesized that the relationship was rather explained by shared pathophysiological pathways in which pain can predispose to depression or anxiety symptoms.”

 

REVIEWER 3

Reviewer #3: Please see attached review comments. Thank you for the opportunity to review this manuscript and to read more about your research. Your research represents a valuable and considerable effort to consider the impact of pain on psychotherapeutic outcomes over a longer-term follow up and I feel that this is important and valid for publication, though the manuscript needs significant further reworking.

Thank you for the opportunity to review this interesting and timely paper. The paper addressed the much-needed area of how pain is related to outcomes of psychotherapeutic interventions. The paper was predominantly well written and added much interesting commentary on returning to work and the authors should be commended for their careful statistical analyses and the length of the follow up to their original pragmatic RCT. Such longitudinal follow-ups are insightful and often uncommon. 

Despite many real strengths in the paper, my recommendation is, unfortunately, a rejection. Principally this is because I hold significant concerns about the validity of the identification of the pain group as a Chronic Spinal Pain group. These are concerns that the authors also note in their limitations section (See Lines 552-557), but do not seem to have mitigated or to considered in greater depth. The authors argue that their analyses are between non-chronic spinal pain and depression, representing a subsample of the original RCT study. However, the rationale for the CSP is, to my mind, quite problematic. They conceptualised the CSP group as people who scored as having some/seriously affected neck, lower or upper back pain and combined this with a question about having musculoskeletal complaints for at least 3 months in a row. Unfortunately, the musculoskeletal complaints selection did not specifically screen for back pain/spinal pain and included a wide range of body areas (inclusive of arm, pelvis, foot, etc.) and therefore it is likely that the sample may represent a huge range of people struggling with pain that may or may not be spinal. For example, people with conditions as varied as chronic pelvic pain, broader chronic pain syndromes or osteoarthritic pain in areas other than the spine could all meet the inclusion criteria. The choice to categorise this as a single chronic spinal pain group seems slightly spurious and feels as though it may not be a robust decision. I would encourage the authors to revisit their inclusion and exclusion criteria and think through the viability of the group inclusion/exclusion criteria, such that they are stronger and more justified. Given this, I am concerned that the validity of the conclusions and analyses and the way in which they are argued in relation to spinal pain are not as strong as is needed for publication. The group does not clearly settle within back pain, nor is it spinal cord injury and the clarity of the inclusion/exclusion is not adequate. 

“This may involve the possibility that participants with SP only during the previous month and with other musculoskeletal pain the previous months were falsely identified as having CSP.” We added: “However, this is not so likely; disabling spinal pain is often non-specific and more frequently accompanied by other musculoskeletal pain than alone standing [ref]. Hence, it may be likely that several of the CSP participants were suffering from other musculoskeletal pain as well.”

We comment on this more later.

Further comments: 

The abstract seems not to have populated fully on the form, losing the conclusion, but this may be because it is a long abstract and would benefit from being reduced, with fewer presentations of statistics and a broader overview presented. The conclusion is, however, present at the start of the manuscript. The abstract seems unusually lengthy. 

The abstract is slightly reduced and the statistics has been removed.

Mental clinical recovery is a slightly unusual choice of terminology. The authors could present this as mental health, mental wellbeing or psychological wellbeing as those phrases would be more familiar to a psychological readership. Clinical recovery is a useful term but is associated with a much broader set of medical implications than is intended in this manuscript. 

We agree with this and has been struggling earlier to define this. Now we have chosen to use the term ‘clinical remission’ and defined it as remission of both depressive and anxiety symptoms at two-years follow-up (CR2yr), i.e. obtaining scores below the clinical cut-off score of both BDI-II (≤ 13) and Beck Anxiety Inventory (BAI, ≤ 9) at 2-year follow-up.

Introduction: 

• Line 52: a reference would be beneficial to evidence your statement on LBP prognosis. 

We included a recent systematic review and a meta-analysis of Wong (2021)

• Grammar and phrasing in lines 53-55 need revisited.

We rephrased and rearranged the order of the lines.

“Focus on the impact of comorbid pain on treatment effectiveness of depression, has been scarce [ref]. Research, however, has mostly focused on the impact of comorbid depression in non-specific low back pain (LBP) interventions. It has been suggested that comorbid depression might have an adverse effect on the prognosis and treatment effectiveness of LBP [ref]. Evidence of a negative impact of the co-existence of depression and chronic LBP on return to work (RTW) is still insufficient[ref].”

• Line 58: effect sizes would be usefully added here such that we can see the extent of the ‘smaller’ size that you mention. 

We have removed this line. Closer look at the meta-analyses we referred to showed us that Brief intervention was compared to mainly no intervention, undescribed, usual (GP) care, waiting lists, or other interventions than psychotherapy. 

• Line 60: a brief overview sentence of your original findings would be useful here for readers unfamiliar with your original paper

We have changed and added this section:

“Even though many studies have favoured short term psychotherapies above brief psychotherapies for depression, we found in a recent pragmatic randomized controlled trial (RCT), that the briefest intervention was superior in enhancing early work participation and long term clinical remission. In the brief psychotherapy, the focus was on normalizing, accepting and coping with their present mental health complaints and hindrance for work participation. In the other intervention, short-term psychotherapy, there was in addition to the themes of the brief intervention focus on processing other challenging issues and previous pathogenic experiences.

Although brief psychotherapy is effective in a heterogeneous population, different intervention responses may occur in subgroups. It is important to identify the subgroups for which brief psychotherapy is insufficient. A considerable part of the participants in our previous study reported both depression and back or neck complaints at baseline.”

• Line 62: it would be helpful to explain here how the focus changed. 

We added to the brief intervention “that focused on normalization”. Further, the interventions are briefly described in the method section.

• Line 64: the word choice of ‘induces’ seems inappropriate here – somatic comorbidity cannot induce psychotherapy sessions.

We have replaced ‘induce’ with ‘entails usually’ “It has been suggested that somatic comorbidity entails usually an increased number of sessions”

• Lines 67-71 detail the hypothesis about why short term psychotherapy may be advantaged over brief psychotherapy, but this seems to directly contrast with the findings from your earlier study that you discuss in line 62. It seems odd to then hypothesise the opposite to what your previous work has suggested. This discrepancy should be resolved or discussed and acknowledged.

We think that treatment of depressed patients with comorbid pain is more challenging as this usually is associated with reduced quality of life and work function. Therefore, short-psychotherapy might be more effective in this subgroup.

We have rephrased the sentence:

“Short term psychotherapy allows for greater focus on previous or current challenging issues, including challenging work and additional pain issues. Short term psychotherapy instead of brief psychotherapy may therefore be more beneficial for the participants with comorbid challenging CSP complaints.” 

• Line 74: the substantial subgroup that is referenced here is unclear – I presume you mean the CSP group, but it would be helpful to be more specific. 

To clarify, we have added the word ‘this’ and ‘with comorbid CSP’:

“Insight into the impact of CSP as a nonspecific treatment predictor or moderating the efficacy of the psychotherapy can be clinically useful in modifying the expectations of intervention efficacy and guide clinical decision-making for this substantial subgroup with comorbid CSP.”

Methods: 

The methods in general did not feel well organised and struggled to read independently as a standalone paper separate from the previous publication. As such, the following recommendations may help the authors clarify this section: 

• Line 87: Please briefly explain how your previous study was a ‘pragmatic’ RCT. 

We have added:

“In order to improve generalisability of findings, a pragmatic design was chosen, following ordinary clinical procedures with respect to patient inclusion, participation in minor additional treatment modalities and decision of therapy termination.”

• Use of hyphens seems unusual throughout the manuscript and hyphens should be checked throughout. 

It is corrected

• Lines 94-99: This section seems to re-read like an abstract, giving results. This is odd here and this section should just factually describe (in brief) the study design and setting. 

We agree, we have removed the concerning sentences.

• Please relist the materials used in the original study and provide a full materials section

The present study applies the same materials as in the original study with Brief Illness Perception questionnaire which we did not discuss there and the exception of Hopkins Symptom Checklist-10. This questionnaire gave overlapping information to the two Beck Inventories (BAI and BDI) and was thus omitted. We hope the reviewer agree that listing the tests once more should not be necessary. In the end of the materials section in the revised manuscript, we now write:

“With the exception of one omitted inventory (Hopkins Symptoms Checklist-10) giving information overlapping with the Beck-inventories and one that was added in these current analyses (BIPQ), the methods applied in the present study equals to those reported in Wormgoor [27]. The difference in methods being that we now look for moderation effects and divides the sample as to CSP or non-CSP.”

• Line 105: it would be more patient-centred to refer to disorders as mental health disorders rather than mental disorders (which implies problematic cognitive function or stigmatising viewpoints). 

We agree and have adjusted this

• In the participants section, please present inclusion criteria before exclusion criteria and clarify the exclusion criteria. What counted as ‘acute or more severe pathology’? It would be useful here to report the processes and judgements that led to exclusion. 

We rearranged the order and added that we followed the ordinary routine criteria:

“The RCT followed the outpatient clinic routine criteria and enrolled patients, aged between 18 and 65 years, who were on or at risk of sick leave, primarily due to common mental health disorders. Sick leave had to be less than nine months during the preceding two years. All referrals were assessed by the clinic’s psychologist coordinator. Participants were included to the clinic irrespective of comorbidities, but in cases of acute or more severe pathology that required greater input than the outpatient clinic could offer, the patient was excluded and referred to other psychiatric service.”

• Why was the cut-off for sick leave 9 months? This feels like an arbitrary figure and no rationale for this cut-off was presented. 

The cut-off 9 months sick leave is the normal procedure of this transdiagnostic outpatient clinic. Longer sick leave dramatically worsen work-return and the patients are thought to need more extensive help than this clinic can offer. Patients are referred to ordinary secondary mental healthcare. As this was a pragmatic trial, we described this as an exclusion criterion.

Intervention: 

• I query the meaningfulness of the quantitative difference between a median of 5 hours in the brief intervention and 7.5 hours in the short intervention - this is only a marginal difference in number of therapeutic hours. This needs further consideration and a rationale/justification for separation as different intervention types. 

We have discussed this more in the discussion in the section ‘implications for psychotherapy’: 

“In both sub-groups, the number of sessions in Short-PsT were considerably lower than aimed at. Although Brief-PsT was a management directive and standard procedure in our clinic, the psychotherapists had aspired to be allowed to expand both focus and number of sessions to improve outcome. Therefore, this substantial reduced applied number of sessions in Short-PsT was unexpected, especially in the CSP subgroup. We are unsure whether spinal pain and related problems had been discussed in particular in the psychotherapy, but there had certainly been the possibility to apply more sessions and address this more thoroughly in the psychotherapy.” 

• What were the decisions around concluding or ending the intervention? Why was there such variation in number of hours within the protocol – the checks around fidelity/manualisation/standardisation of the intervention delivery are not clear in this manuscript (though I accept that this may not always be possible within pure psychotherapy). Further detail and consideration is needed about description of the intervention and the associated screening/delivery/termination. 

As this was a pragmatic trial, the psychotherapy was not manualised and decisions of therapy termination were done cooperatively by the psychotherapist and patient, usually because of remission or lack of improvement.

Measures: 

This should be factually presented either as a materials section (as mentioned above), where questionnaires included in the study are detailed here each in turn. Or could be presented as a primary and secondary outcome measures section. It is unusual to have DVs and IVs presented here – these might be better located in the design section in a much briefer form. Indeed at the end of your manuscript you refer to your ‘secondary outcome measures’ on line 564, but we don’t know what these are from your current presentation of the manuscript. 

We removed the term ‘secondary outcome measures.

In general, there are a huge number of measures in this study which are contained in various sections across the method and it is difficult to see in a coherent way what was tested, on which scales and the reliability/validity of those measures and subscales. Reliability and validity of the measures in your study and how they compare with expected clinical reliability/validity checks should be presented. 

We admit that there is a huge number of measures. Since the original study reported the same huge amount of measures, we were concerned that the present study should use the same measures, which we think is your concern also when asking us to list the methods used in the original paper. Reviewer 2 is concerned that we may have omitted potential important covariates when we decide only to continue analysing those variables where the CSP and non-CSP groups differed with a significance level of .15. 

All questionnaires we used here are in common use in the field of common spinal pain and mental health complaints, in clinical and research settings. Validity and reliability of Norwegian versions have been evaluated. Unfortunately, we have not discussed this in particular.

Please present a Statistics and Data Analysis Section in which you fully detail your data management methods, what you did to mitigate against missing data, and your analytic strategy. The ‘potential covariates’ section is unusual and dense, containing much information that should be in the materials section (e.g. scoring of the BAI) and information about how you ran the study (which should be in the procedure section). This should be collapsed with your statistical analyses section and some significant care taken to present data management and analysis decisions clearly and concisely in a single location within the manuscript.

We have reorganised, rewritten and shortened the ‘Measures’ section

We changed the section heading ‘Statistical analyses’ to ‘Statistics and data analysis’. 

Line 172: please present some statistical validation for the use of the adapted version – does the internal consistency match the original measure?

We are not discussing this or statistical validated this in detail, only included a reference and commented the following:

“The FABQ, although originally developed for LBP, adapted versions has been evaluated and widely used for other populations and considered as a useful prognostic tool for individuals on sick leave due to both musculoskeletal and psychological disorders[38, 39]” 

Why are particular questions selected for inclusion in analysis rather than using full questionnaires? E.g. line 192? The selection of individual questions raises concerns about the validity and robustness of the data when repeatedly treated in isolation from the measures as a whole.

The questions you are referring to are part of the assessment of work related factors in the primary trial and in there also applied as single items. Several of these questions were also in use in the standard assessment at start of the treatment. The original questionnaires are quit lengthy (22, 17 and 122 items), and too extensive for our assessment of work related factors.

A huge number of outcome measures were treated dichotomously rather than as interval variables. The decisions around where the dichotomous split was chosen to be located seems random and no justification is given or checks about the appropriateness of this. Indeed Line 213 states variables were dichotomized according to the median or other ‘pragmatic decisions’ but these are not detailed. It is not appropriate just to opt to create splits simply for pragmatism. Referenced and robust decisions should inform such choices. I would recommend that, where possible, creation of dichotomous variables should be avoided and the authors should consider alternative methods of analysis, including Bayesian statistics (to account for small subgroups) or bootstrapping. 

We have reduced the number of dichotomous variables and included several variables at interval scale. 

Line 187: please use quotation marks for “my job is… 

OK

Statistical Analyses: 

Lines 196-197: why are the hypotheses repeated here? This is not needed. 

They are removed

Line 198: what is meant by baseline variables were ‘participanted’?

It should have been ‘subjected’, but a ‘find-and-replace action’ has changed ‘subject’ to ‘participant’ by mistake. It is corrected.

Lines 201-204: please report the data check outcomes. Which measures failed and on which tests? What was done about missing data? Was missing data MNAR or MCAR? Little’s test would be useful here.

Here, missing data of clinical remission was more disturbing than the minimal missing data of the baseline data. At start of the results, we have checked whether loss-to-follow-up was related to any of the baseline variables and reported this. In the comparison of work participation degree between the two subgroups, we report now full ITT analyses instead of partial-ITT with missing data replaced by the last-observation-carried-forward rule.

In addition, we added a line concerning missing baseline data: 

“Missing values of the applied baseline values were, apart from the work characteristics items, close to zero percent and completely at random (Little’s test, p= .24).”

In the ‘Statistics and Data Analysis’ section, we added: “Little's missing completely at random test was applied to explore relevant baseline data. Subgroup differences in loss-to-follow-up for WP2yr and CR2yr were tested for all baseline variables. All analysis were performed by original assigned intervention groups. For the presentation of WP-state at follow-up, full intention-to-treat (ITT) analyses were applied; the last-observation-carried-forward rule (LOCF) for missing data was used (last observation could be baseline, 3 months or 1-year follow-up). For the analysis with WP2yr and CR2yr, he data were analysed as partial-intention-to-treat analyses (PITT), excluding participants with no follow-up data.”

What is the rationale for the prediction models? How are these theoretically-informed? It would be useful to justify these models more clearly in the literature that you review in the introduction. 

We have improved the introduction. 

In general, crude odds ratios should be avoided if possible and authors should aim to prioritise adjusted odds ratios. 

We think that both crude and adjusted ratios (cOR and aOR) are important to report and therefore we presented both. In our case, the mentioned cOR will estimate the relative risk of positive treatment response between the patients with and without CSP. This gives us a general expectation of treatment response for the CSP patients we meet in clinical practice. aOR, however, helps us to understand how other variables affects this relation and why less patients obtain good outcome. 

Here we see that it is not necessarily having substantial spinal pain that affects treatment response, but in case of clinical remission, we see it is rather the fact that many of these patients have high anxiety scores at baseline as well. In case of work participation as outcome, it shows that both CSP and previous sick leave duration (which was higher in the CSP group) are important determinants.

Results: 

Effect sizes and confidence intervals should be presented throughout – these are not consistently presented. Please ensure they are present either in tables or throughout the manuscript. 

We have added effect sizes and confidence intervals in the tables if relevant.

Table 1: 95% CI or IQR

Table 2. Effect size: OR or Cramer’s V

Table 3. 95% CI and OR

Table 6. 95% CI or IQR

Table 7. 95% CI and OR

Please italicise p-values. 

It is corrected.

Lines 280-282: Grammar seems unusual in this sentence. 

We rephrased these lines:

“At baseline assessment, the two subgroups showed comparable WP rates. However, in the previous year, CSP patients had more sick leave related to their current health problems. CSP patients experienced more often their job as physically demanding, yet a higher proportions indicated that they liked being at work.”

The language in the results section can sometimes be tentative, suggesting that the authors are underconfident about stating the results strongly. E.g. line 341 ‘seemed to predict’ or 334 ‘seemed thus not a significant dilemma’. Please check throughout.

We removed many ‘seemed’ words.

Discussion: 

Line 381: avoid the contraction of did not.

It is corrected.

Throughout the discussion, the authors restate statistical findings and use statistical language. I would suggest that the authors revisit the discussion section in full and aim to avoid re-presentation of statistical findings and instead focus on the (clinical) implications of their findings and their relationship with psychological/medical literature and psychological theory. By reducing the amount of time spent revisiting the statistics, there would be greater space for linking the findings with previous literature, which would be more informative for the reader and would locate the manuscript more strongly within the literature base.

We agree, and have worked through the Discussion section and omitted almost all repetitions of numbers from the Results section. Instead, we could then highlight the phenomena and discuss them, as suggested, to what others have found.

It would be useful to have further elaboration of the mechanisms and theoretical justification for the findings, particularly around lines 427 and 428. There is minimal inclusion of theory-driven explanation in the manuscript at present. 

We have tried to discuss possible mechanisms and theoretical issues concerning our findings. We have not discuss the indicated lines more in particular “In spite of this physical work load, their level of fear avoidance beliefs related to work was comparable as well, and they reported higher job satisfaction. The latter may have turned their sick-leave expectations more positive.”

Lines 459-460. This is an interesting finding and it would be helpful here to have more discussion about why correlations with depression but not anxiety might have occurred. 

Referring to: “It was therefore remarkable that the presence of SP at follow-up was only correlated with current clinical depression and not with anxiety.”

Unfortunately, this finding have been removed from results (former Table 2) and discussion as part of our attempt to simplify this table and to remove data that was less relevant to our aim and hypotheses. SP at follow-up time was not (part of) an outcome response measure.

Line 467: check grammar in ‘worth to consider’ (worth considering?)

It is corrected.

Line 492: if you are presenting the lack of association as interesting, it would be great to see more speculation/literature-based explanation about this rather than suggesting it is out of scope.

“The lack of association between WP2yr and CR2yr in non-CSP is interesting, but behind the scope of this paper”

Instead, we wrote: “The lack of association between WP2yr and CR2yr in non-CSP is interesting. Both interventions seemed effective in the non-CSP participants for acquiring acceptance that subjective health complaints are normal features of human life and consequential assurance that continuing normal life and being at work may be crucial in managing these. In the CSP participants this seemed less successful; perhaps the additional burden of having pain issues are more difficult to handle at work, especially in light of the higher reported physical work load.”

Lines 506-512: It is possible that the moderating impact of CSP on psychotherapy could be muted by the lack of differentiation between the intervention types/hours and also by the lack of clarity of the recruitment strategies. This needs discussed. 

We considered the two psychotherapies as quit different. The number of psychotherapy sessions was significantly higher, although in some other settings median 5 and 8 sessions (in CSP) are both regarded as brief. However, we believe that the difference in focus was the most significant difference.

In the primary study, both early work participation and long term clinical remission were clearly higher in the Brief compared to the Short psychotherapy. In our primary paper, we discussed whether it is possible that in Short-PsT the additional focus on previous or current challenging issues confounded the process of acceptance of and coping with common health complaints. Even though the aim of Short-PsT was to resolve past and current psychological issues, one may ask whether the additional focus to a certain extent also led to perpetuated worrying, ruminating and focusing on symptoms.

The lack of statistical significance, we think, was rather a power problem. We had only 23 and 28 patients for analysing the clinical remission in CSP. 

Lines 519-522: Pain acceptance is not typically seen as a physiotherapy issue. This is more usually dealt with in Acceptance and Commitment therapy which is psychology led. Therefore I don’t feel that the physio consult would have induced acceptance and coping, so this section feels weak. 

In modern treatment of long-lasting spinal pain, the main aim is not reduction of pain, but improvement of functioning. For this, acceptance and coping of the long-lasting symptoms is essential. A thorough physical examination will assure the patient that nothing is seriously wrong (which usually is the case), explain findings and consistently focus on the overall message that the spine is strong. In addition to evidence based information, practical advice and motivating and encouraging the patient to stay active despite the pain, the patient will attain coping skills to improve accept and manage their spinal pain.

We have strengthen the general focus of acceptance of general common health complaints somewhat in the intervention section:

“The study was incorporated into the usual healthcare setting. Their general approach was the assumption that subjective health complaints are inevitable features of human life and providing the patients with coping skills to manage these. Before starting psychotherapy patients participated in a 5-hours transdiagnostic group education. The education was aimed at providing insight and understanding with common health complaints, including musculoskeletal pain, anxiety, depression, and stress. 

Following the education, concurrently to the psychotherapy, all participants had the opportunity to participate in a five-day coping course, individual coaching sessions with another health care professional or clinical examination with a physiotherapist.” 

In the Discussion we have added:

“They could participate on their own initiative or on their psychotherapist’s recommendation.”

It may be also useful to think through the extent to which your results may reflect the fact that there could be an issue here in terms of stratification of patients to the ‘best’ treatment approach. For example, it is possible that the ‘CSP’ group may have shown better outcomes if they were first seen within pain management services rather than in talking therapies. This research may suggest that this stratification/screening should occur. 

For all patients included in the study, mental health issues were seen as their main issue. However, nearly all patients participated to a transdiagnostic education group, prior to psychotherapy, but after baseline assessment and randomisation. This education focussed on managing pain issues as well. Consultation of a physiotherapist was possible whenever desired in their treatment course.

In the Intervention section and discussion mentioned above, we have shortly described the availability of ‘pain management services’ 

In the Discussion we mentioned already: “In the future, for the patients with spinal pain we may consider to put more emphasize on the value of a physiotherapist consult prior to a referral for psychotherapy.”

Line 552: I strongly agree with this comment and feel this problematises the manuscript as a whole. 

We are fully aware of that the CSP group may include patients with other musculoskeletal pain complaints as well. This reflects common clinical practice as well. Their might be a theoretical possibility that patients were included that had other long-lasting (≥3 months) musculoskeletal pain and only substantial spinal pain in the last month. Still we think these patients are relevant to include in the CSP group as well.

We have added: “The participants that reported substantial spinal pain during the preceding months and had a current musculoskeletal pain episode of at least three months at baseline assessment, were labelled as CSP. This may involve the possibility that participants with spinal pain only during the previous month and with other musculoskeletal pain the previous months were falsely identified as having CSP. However, this is not so likely; disabling spinal pain is often non-specific and more frequently accompanied by other musculoskeletal pain than alone standing[ref]. Hence, it may be likely that several of the CSP participants were suffering from other musculoskeletal pain as well.”

In addition, we performed sensitivity analyses to see how redefining the threshold for subgroup defining changes the observed subgroup and intervention for differences for WP2yr and CR2yr. We tested four different thresholds. For all variants, contrasts and level of significance between subgroup outcome of WP2yr and CR2yr was greatest in the subgrouping that is reported here.

Conclusion: It would be helpful in the final paragraph (586-588) to have an additional sentence about the meaning of the fact that the impact could not be confirmed in the current study. An explanation of the importance of a null funding is of equal importance as the findings that were positive. 

We have changed the Conclusion. 

“The literature regarding tight associations between anxiety, depression and pain is overwhelming, telling us that mental health affects pain and pain affects mental health. Although this calls for psychotherapy as possible treatment, the present study corroborate previous findings in that subjects with spinal pain, anxiety and depression profit less from psychotherapy than subjects with anxiety and depression alone. Presence of CSP results in less work participation two years after the therapy and worse remission of mental health symptoms as well. The association of older age and of more severe baseline anxiety are of respective relevance for this reduced effectiveness. The expectation that more severe problems demand more extensive therapy, i.e. that subjects with comorbid CSP would profit more from a lengthier therapy process was not confirmed. Both the brief and the short-therapy format was equally less effective in treating comorbid CSP. As the length and focus of therapy, i.e. either acceptance and normalizing or additionally addressing psychologically central themes, did not mediate effectivity in treating comorbid CSP. Improvement of therapy for the CSP group must probably be sought by trying out other techniques.”

---

## [Decision Letter · Decision Letter 1]

30 Mar 2022

PONE-D-20-20567R1The impact of comorbid spinal pain in depression on work-participation and clinical remission following brief or short psychotherapy. Secondary analysis of a randomized controlled trial with two-year follow-upPLOS ONE

Dear Dr. Wormgoor,

Thank you for submitting your manuscript to PLOS ONE. After careful consideration, we feel that it has merit but does not fully meet PLOS ONE’s publication criteria as it currently stands. Therefore, we invite you to submit a revised version of the manuscript that addresses the points raised during the review process.

We look forward to receiving your revised manuscript.

Kind regards,

Vanessa Carels

Staff Editor

PLOS ONE

Reviewers' comments:

Reviewer's Responses to Questions

**Comments to the Author**

1. If the authors have adequately addressed your comments raised in a previous round of review and you feel that this manuscript is now acceptable for publication, you may indicate that here to bypass the “Comments to the Author” section, enter your conflict of interest statement in the “Confidential to Editor” section, and submit your "Accept" recommendation.

Reviewer #1: All comments have been addressed

Reviewer #2: (No Response)

2. Is the manuscript technically sound, and do the data support the conclusions?

Reviewer #1: Partly

Reviewer #2: Partly

3. Has the statistical analysis been performed appropriately and rigorously? 

Reviewer #1: Yes

Reviewer #2: Yes

4. Have the authors made all data underlying the findings in their manuscript fully available?

Reviewer #1: Yes

Reviewer #2: No

5. Is the manuscript presented in an intelligible fashion and written in standard English?

Reviewer #1: Yes

Reviewer #2: Yes

6. Review Comments to the Author

Reviewer #1: The authors have put in great effort to address the comments. The authors have provided detailed information and clarity on the comments. I have no further comments.

Reviewer #2: Review of paper:

The impact of comorbid spinal pain in depression on work-partcipation and clinical remission following brief or short psychotherapy. Secondary analysis of a randomized controlled trial with two-year follow-up

Abstract:

The abstract is more readable, however, you should keep the estimates in the result section (not delete them as suggested in the revised manuscript). As previously highlighted, there is still inconsistency between the aim of the study and the conclusion.

Introduction:

Line 112-117: Suggest that you describe the aim of the study as research questions rather than hypothesis-testing.

Method:

Line 130-131: Not necessary to write the aim in this section (already described). Here you also write “primary” aim. If you have a primary and secondary aim, this should be described clearly.

Line 280: Not clear what you mean by “potential baseline covariates”. These are descriptive variables, and not variables that you want to include in statistical analysis according to your aim..

Line 289-292: It is not clear how the described statistical tests comparing the subgroups are needed in relation to your aim. And what it add to the conclusion of the study. Suggest to remove this.

Line- 298-231: I would not recommend to base the inclusion of possible confounders in the model on statistical tests, rather base this on a clinical and theoretical foundation.

Results:

Line 362-375: Can this section be shortened?

Line 377 -> : This section is not specified as an aim, nether as suggested hypothesis, and I don’t see the relevance in performing al these statistical tests. Suggest to remove or rephrase the aim of the study.

Table 4: Still not related to the aim of your study. You only describe in the aim that you are interested in the predictive ability of CSP, and not anything about identifying possible prognostic factors in general. This should be aligned. Or the information/results could be put in supplementary files.

Discussion:

Section from 550 ->: This section can be shortened, as it is not the main aim of you study to identify the differences between participants with and without CSP.

Line 656-665: Text needs to be written more clear, and better relate it to the context of your study.

Line 680-682: You argue that fear avoidance are equally distributed in both subgroups, but you report the baseline value as median. Is it not possible that the individual variation within the subgroup still may influence the association?

Limitations: You still don’t thoroughly describe how the limitation in the study may influence the specific results of the analyses. A discussion of the sensitivity analyses can for instance be relevant here..

Conclusion: You highlight the research question regarding the short vs. long intervention. This should also be more clear in the aim of the study.

7. PLOS authors have the option to publish the peer review history of their article (what does this mean?). If published, this will include your full peer review and any attached files.

Reviewer #1: No

Reviewer #2: No

---

## [Author Response · Author response to Decision Letter 1]

19 May 2022

Dear editor and reviewer,

We appreciate the time and effort that you have dedicated to providing valuable feedback on our manuscript. We have considered all comments and incorporated changes and/or motivations to reflect most of the suggestions provided. 

We see that the reviewer comments were related to three main concerns: 

• First, our attention to comparing the two subgroups at baseline and to apply this to identify relevant covariates (confounders) as (s)he indicated earlier as well. We have tried to improve the reasoning for this even more. 

• The second concern was the additional aim of the current study to evaluate whether the occurrence of comorbid spinal pain brought about effect modifications following the two different interventions. We concluded on this, but apparently, we had not made this clear enough as a defined aim of the study. 

• Third, the reviewer suggested to remove, shorten or undo some removals of paragraphs in the revised manuscript. However, this mainly concerned suggestions from the other reviewers and we have chosen to keep with these in case we considered these as improvements to our first manuscript.

Throughout the manuscript we strived to me more clear by using correct and consistent terminology. When aiming at the impact of comorbid spinal pain on long-term outcome we focus on CSP as a ‘prognostic factor’. In the evaluation whether the two interventions has differential effect on the patients with or without comorbid spinal pain we use the term ‘effect modification’. During the process of selecting other baseline characteristics as ‘covariates’ we label these as ‘potential cofounders’.

Below are comments to the suggestions of reviewer 2 and a point by point account of changes made in the revised manuscript. The black text in italics are the comments by the reviewer while our comments are in blue and the changes in the manuscript are marked in green.

Abstract

The abstract is more readable, however, you should keep the estimates in the result section (not delete them as suggested in the revised manuscript). As previously highlighted, there is still inconsistency between the aim of the study and the conclusion.

The estimates had been removed after earlier suggestion of another reviewer to reduce presentations of statistics and rather present a broader overview. We have kept us to this, especially because of the number of words in the abstract.

We apologize for not taking this issue seriously enough in the former revision and have improved the abstract, particularly ‘objectives and ‘conclusions’:

“Objectives: This explorative study analyses the influence of baseline comorbid long-lasting spinal pain (CSP) on improvement of long term work participation and clinical remission of mental health illness following either brief coping-focussed or short-term psychotherapy for depression. Whether type of treatment modify outcome with or without CSP is also analysed. 

“Conclusions: CSP at baseline reduced work participation and worsened remission of mental health symptoms two-year following psychotherapy. Older age and more severe baseline anxiety are associated to reduced effectiveness. Type of psychotherapy received did not contribute to differences in intervention effects.”

Introduction

Line 112-117: Suggest that you describe the aim of the study as research questions rather than hypothesis-testing.

We have rewritten the aim of the study as research questions and removed our hypotheses and objectives:

“The main research question of the current study is to determine whether baseline CSP is prognostic for long-term sick leave and reduced clinical remission of mental health problems following psychotherapy for depression. Second, it is questioned whether CSP may moderate the effect of brief versus short psychotherapy. To answer the first question, we will evaluate whether CSP is an effect modifier for improvements on sick leave and clinical remission of mental health problems two-year following psychotherapy for depression and to estimate the magnitude of that effect, if any. To enable adjustment for confounding, we need to examine the differential associations between CSP and other baseline characteristics and their association with intervention outcome as well. Secondly, we will evaluate whether the effects of brief-coping focused and short psychotherapy with a more extended focus are equally beneficial in participants with baseline CSP compared to participants without CSP, or not.”

Method

Line 130-131: Not necessary to write the aim in this section (already described). Here you also write “primary” aim. If you have a primary and secondary aim, this should be described clearly.

The primary and secondary aim mentioned in this section referred to the aims of the primary RCT. We acknowledge that this can be confusing and removed this sentence. 

Line 280: Not clear what you mean by “potential baseline covariates”. These are descriptive variables, and not variables that you want to include in statistical analysis according to your aim

Line 289-292: It is not clear how the described statistical tests comparing the subgroups are needed in relation to your aim. And what it add to the conclusion of the study. Suggest to remove this.

Line- 298-231: I would not recommend to base the inclusion of possible confounders in the model on statistical tests, rather base this on a clinical and theoretical foundation.

As our focus is on the impact of CSP on intervention outcome we need to adjust for possible confounding baseline variables. These confounders are the variables that are both related to baseline CSP and affect intervention outcome and we need to include those in our model as covariates. The descriptive variables were pre-selected for their expected potential of being a confounder based on literature review and our clinical experience.

In the process of selecting relevant covariates as described in step 1 and 2 of the development of the prediction model, we had labelled them as ‘potential baseline covariates’. However, in this revised manuscript we have tried to make it more clear why it is essential to perform some analysis with these to evaluate the statistical associations of these descriptive variables with both the dependent variable CPS (step 1) and with the independent intervention effect variables (step 2) as well. We therefore changed the wording from “potential baseline covariates” to “potential confounding baseline variables” to make it more precise what the actual role of the covariates in the model are.

“Potential confounding baseline variables were selected based on a review of the literature concerning work participation and clinical outcome in common mental health or spinal complaints (table 1). In this context, confounding variables are both related to CSP and affect intervention outcome. The included baseline variables were subjected to descriptive statistical analysis stratified for the presence of CSP. Data of both intervention arms were pooled. Their relationship with both CSP and intervention outcome were evaluated as described below in the selection of covariates for the regression models.”

To examine the prognostic and effect modifying impact of CSP, two separate models were tested, one for each treatment response variable: WP2yr and CR2yr. Each prediction model was developed in five consecutive steps. Step 1 involved selection of potential confounders as covariates, based on the results of the association of the initial selected baseline variables with CSP. To be comprehensive we chose a lenient level of differential significance for including possible covariates (p.15). In step 2, analyses (Chi-squared test with Yates’ continuity correction for all 2x2 analyses and Mann–Whitney U test for numerical variables) were applied to assess the associations of these variables with baseline WP and treatment response for both defined binary dependent outcome variables: WP2yr and CR2yr.

Results

Line 362-375: Can this section be shortened?

We acknowledge that you commented on this same section in your previous review. Unfortunately, we did not shorten this section as a compromise with one of the other reviewers who requested even more information. We think it is informative.

Line 377 -> : This section is not specified as an aim, nether as suggested hypothesis, and I don’t see the relevance in performing al these statistical tests. Suggest to remove or rephrase the aim of the study.

Table 4: Still not related to the aim of your study. You only describe in the aim that you are interested in the predictive ability of CSP, and not anything about identifying possible prognostic factors in general. This should be aligned. Or the information/results could be put in supplementary files.

I think we have specified the importance of including this section and the statistics above. In the description of the study aim we included “To enable adjustment for confounding, we need to examine the differential associations between CSP and other baseline characteristics and their association with intervention outcome as well.” 

And we start the concerning section ‘Comparative characteristics according to the presence of comorbid spinal pain’ with “To assess the balance between the groups at baseline with respect to the main prognostic or confounding factors, relevant baseline measures stratified for the presence of spinal pain (subgroup non-CSP vs subgroup CSP) are presented in Table 1.”

Discussion

Section from 550 ->: This section can be shortened, as it is not the main aim of you study to identify the differences between participants with and without CSP.

As mentioned previously, we think this section is strongly relevant to the main aim of the study to identify the specific effect of CSP on intervention effect. In this, it is necessary to assess what the relevance is of the differential characteristics of this subpopulation. We have added:

“Confounding is by definition a nonissue in RCTs, but in the comparison of intervention effects within subgroups of a RCT it is essential to assess possible confounding factors that are related to both the occurrence of spinal pain and the intervention outcome measures.”

Line 656-665: Text needs to be written more clear, and better relate it to the context of your study.

We have rewritten this paragraph and hope it will be more clear and related to the study. The new test read: 

The present findings of higher baseline anxiety in CSP, rendering the impact of CSP on symptom remission as insignificant, is complementary to previous research reporting negative prognostic value of baseline anxiety on pain outcome [76, 83, 85]. The causative mechanism behind the association is not clear and may include several pathways. Reviewing this, Gureje [60] claims that the pattern of association supports a causal pathway that proceeds from mental disorder to chronic pain rather than the reverse. In a longitudinal study among individual with chronic pain it was found, indeed, that neither pain nor pain-related disability predicted depression/anxiety [86]. Conversely, symptoms of both depression and anxiety, prospectively predicted levels of pain and pain related disability. However, findings from a large longitudinal twin study [87] found that pain predisposed for depression or anxiety symptoms, indicating a relationship explained by shared pathophysiological pathways. Possibly, the coexistence of anxiety and pain may point to a reciprocal relationship in which the direction might be dependent on context. Negative health cognitive bias, shared neurobiological features and somatization processes may mediate this relationship [86]. Based on the results of our present study, we will limit ourselves to informing clinicians that pain and anxiety may covaries and that both would have to be addressed in therapy.

Line 680-682: You argue that fear avoidance are equally distributed in both subgroups, but you report the baseline value as median. Is it not possible that the individual variation within the subgroup still may influence the association?

We agree that fear avoidance behaviour in principle may be an obvious differential variable comparing the subgroups with and without CSP. As higher scores indicate that the individual has elevated fear-avoidance beliefs towards work, we may expect higher FABQ-work scores in the CSP group because of the additional burden of having CSP. In general, a total FABQ-work score higher than mean 4.85 is considered as a high, positive FABQ-test result.

We interpret your comment to suggest that the median may be misleading in the sense that the within group distribution may still vary so much that a group difference is actually present, but not visible by comparing the median performance of the two groups. The median score presented here is the median of the individual mean scores of all FABQ-w items. However, we have also reported (see table 1) the interquartile ranges of this measures that showed quite similar variance within the subgroups (for FABQ-w 1.5-4.1 vs. 1.1-3.9). In the analysis comparing the median, it is important to stress that there were neither any close to significant tendency for group differences with regard to fear avoidance (p=.35). Thus, we think it is defendable to state in the Discussion part that there were no differences.

Limitations: You still don’t thoroughly describe how the limitation in the study may influence the specific results of the analyses. A discussion of the sensitivity analyses can for instance be relevant here.

We have included in the limitation section:

“Although subgrouping was based on a pre-randomized baseline characteristic, a point of concern is that the decision to perform subgroup analyses was post hoc. The RCT was therefore possibly not adequately powered to do so. Consequently, the probability of generating ‘false-negative’ results in the analyses in this current study are substantial.” 

…. “Although sensitivity analysis for subgroup are especially relevant for post-randomization groups, we did however, perform sensitivity analyses to see how redefining the threshold for subgroup defining changed the observed subgroup and intervention outcome differences. For all four variants, contrasts and level of significance between subgroup outcome was greatest in the subgrouping that is reported here.”

Conclusion

You highlight the research question regarding the short vs. long intervention. This should also be more clear in the aim of the study.

We hope it is clearer now from the defined research questions that the study aims at both assessing the influence of baseline comorbid long-lasting spinal pain (CSP) on intervention improvements and in addition at examining the possible treatment effect modification.

Thank you for giving us the opportunity to once more revising our manuscript.

Best regards

Marjon Wormgoor

---

## [Decision Letter · Decision Letter 2]

5 Aug 2022

The impact of comorbid spinal pain in depression on work-participation and clinical remission following brief or short psychotherapy. Secondary analysis of a randomized controlled trial with two-year follow-up

PONE-D-20-20567R2

Dear Dr. Wormgoor,

We’re pleased to inform you that your manuscript has been judged scientifically suitable for publication and will be formally accepted for publication once it meets all outstanding technical requirements.

Kind regards,

George Vousden

Staff Editor

PLOS ONE

Additional Editor Comments (optional):

Reviewers' comments:

Reviewer's Responses to Questions

**Comments to the Author**

1. If the authors have adequately addressed your comments raised in a previous round of review and you feel that this manuscript is now acceptable for publication, you may indicate that here to bypass the “Comments to the Author” section, enter your conflict of interest statement in the “Confidential to Editor” section, and submit your "Accept" recommendation.

Reviewer #1: All comments have been addressed

2. Is the manuscript technically sound, and do the data support the conclusions?

Reviewer #1: (No Response)

3. Has the statistical analysis been performed appropriately and rigorously? 

Reviewer #1: (No Response)

4. Have the authors made all data underlying the findings in their manuscript fully available?

Reviewer #1: (No Response)

5. Is the manuscript presented in an intelligible fashion and written in standard English?

Reviewer #1: (No Response)

6. Review Comments to the Author

Reviewer #1: (No Response)

7. PLOS authors have the option to publish the peer review history of their article (what does this mean?). If published, this will include your full peer review and any attached files.

Reviewer #1: No

---

## [Editor Report · Acceptance letter]

11 Aug 2022

PONE-D-20-20567R2 

The impact of comorbid spinal pain in depression on work participation and clinical remission following brief or short psychotherapy. Secondary analysis of a randomized controlled trial with two-year follow-up 

Dear Dr. Wormgoor:

I'm pleased to inform you that your manuscript has been deemed suitable for publication in PLOS ONE. Congratulations! Your manuscript is now with our production department. 

Kind regards, 

on behalf of

Dr. George Vousden 

Staff Editor

PLOS ONE